# MIXTURE OF MASTERS: SPARSE CHESS LANGUAGE MODELS WITH PLAYER ROUTING

## ABSTRACT

Modern chess language models are dense transformers trained on millions of games played by thousands of high-rated individuals. However, these monolithic networks tend to collapse into mode-averaged behavior, where stylistic boundaries are blurred, and rare but effective strategies are suppressed. To counteract homogenization, we introduce MIXTURE-OF-MASTERS (MOM), the first chess mixture-of-experts model with small-sized GPT experts emulating world-class grandmasters. Each expert is trained with a combination of self-supervised learning and reinforcement learning guided by chess-specific rewards. For each move, a post-hoc learnable gating network selects the most appropriate persona to channel depending on the game state, allowing MOM to switch its style dynamically—e.g., Tal's offensive vocation, Capablanca's positional dominance, or Petrosian's defensive solidity. To quantitatively assess whether each expert captures a distinctive playing signature, we propose a behavioral stylometry metric by training a vision transformer encoder to classify grandmasters from game segments. When evaluated against Stockfish on unseen standard games, MOM outperforms both dense individual expert networks and popular GPT baselines trained on aggregated data.

## 1 INTRODUCTION

Originating nearly 1,500 years ago, chess ranks among the oldest and most thoroughly studied board games. With a game-tree complexity of $\sim 10^{120}$ (Shannon number)–far exceeding the number of estimated atoms in the observable universe–it demands strategic planning and creative thinking. AI surpassed human chess capability roughly two decades ago, beginning with IBM's Deep Blue defeating world champion Garry Kasparov in 1997 using specialized hardware and tree search algorithms (Campbell et al., 2002). DeepMind's AlphaZero revolutionized the field in 2017 by removing human input through reinforcement learning (RL) and self-play (Silver et al., 2017), inspiring contemporary engines such as Stockfish and Leela Chess Zero to adopt neural-network evaluation (Klein, 2022; Maharaj et al., 2022). The latest paradigm shift reframes chess as a language modeling problem, with transformer-based models learning rules and patterns from game transcripts in algebraic notation without explicit search mechanisms (Karvonen, 2024; Toshniwal et al., 2022).

Chess history teaches us there is no single optimal way to play—champions with contrasting styles (e.g., positional, tactical, defensive) have all succeeded at the highest level (Kasparov, 2003–2006). In particular, creativity is a hallmark of chess excellence, embodying the ability to find unexpected, unconventional, yet valid moves that defy standard patterns. However, contemporary chess language models are led by monolithic architectures that struggle with creative play. A dense model, trained to minimize error across billions of moves from millions of players, might hesitate to choose rare or eccentric lines, preferring safe options that conform to dataset statistics. This carries the risk of strategic conservatism and stylistic flattening: the unique traits of individual players may get diluted into a generic behavior. Many chess professionals have warned against a possible homogenization of play with the widespread adoption of AI (Alimpic, 2024; Barrish et al., 2023). We further substantiate these concerns through a survey of expert student and faculty players from 18 universities in 10 countries and 3 continents, detailed in Appendix A. If everyone studies the same moves recommended by the engines, the players may adopt similar strategies and openings, reducing the diversity of ideas in the games. This concern mirrors the trends observed in text generation, where studies indicate that the use of large language models (LLMs) results in a decline in expressive di-

versity, with writing styles converging toward dominant expressions while less common traits are suppressed (Padmakumar & He, 2024; Sourati et al., 2025).

Recent developments suggest that sparse and modular mixture-of-experts (MoE) models may hold promise in computer chess (Helfenstein et al., 2024). From 2021 to 2025, MoE architectures have undergone significant evolution, progressively redefining the notion of "experts"— shifting from feed-forward layers (Fedus et al., 2022; Jiang et al., 2024; Lepikhin et al., 2021) to adapters (Muqeeth et al., 2024; Wu et al., 2024) and full-model branches (Simonds et al., 2024; Zhang et al., 2025a). As the complexity and expressiveness of experts have increased, a natural question arises: *Can we envision a persona-based MoE for chess?*

Building on this reasoning, we introduce MIXTURE-OF-MASTERS (MoM), the first chess MoE with experts emulating world-class grandmasters (GMs). We train multiple small-scale GPT models independently, each on the games of a specific GM, preserving their distinctive styles without cross-contamination (§ 3.1). These specialized models are then combined into a unified sparse language model following a "wisdom of the crowd" paradigm (§ 3.2). A gating mechanism determines which GM to consult for next move prediction[1] depending on the game state. Therefore, MoM acts as a coalition of renowned players, each contributing their situational insight to the evolving board. This approach is attractive for several reasons. First, MoM can prevent collapse toward the majority style and better handle out-of-distribution positions. Second, it provides greater interpretability, enabling analysts to trace decisions back to identifiable chess personas. Third, the modular architecture creates educational opportunities through configurable opponents, facilitating targeted improvement. We introduce a novel model-based metric for behavioral stylometry (§ 3.3), and evaluate MoM on unseen standard games with rich ablations (§ 4). Quantitatively, MoM outperforms single experts and dense baselines trained on games authored by millions of players.

## 2 RELATED WORK

Our work advances the paradigm of chess language models, which has recently seen transformers achieve high performance either through self-supervised learning (SSL) (Karvonen, 2024) or the supervised distillation of state-of-the-art engines to grade candidate moves (Ruoss et al., 2024; Monroe & Team, 2024). In contrast, we employ a hybrid training scheme of SSL and RL to develop a lightweight and controllable MoE architecture where each expert is specialized to emulate a specific human player's style. This player-centric specialization differs from prior chess MoEs that partitioned expertise by game phase (Helfenstein et al., 2024). Finally, we contribute a novel form of behavioral stylometry; while existing models for player identification operate on symbolic move data with human-engineered features (McIlroy-Young et al., 2021; 2022), we operate on raw game video recordings directly. See extended discussion in Appendix B.

## 3 METHOD

Inspired by Li et al. (2022); Zhang et al. (2025a), our sparse MoM model–depicted conceptually in Figure 1–is constructed in three stages.

1. **Branch.** We create $|\mathcal{E}|$ expert replicas $\mathcal{E} = (\varepsilon_{\phi_1}, \ldots, \varepsilon_{\phi_{|\mathcal{E}|}})$ of a $\phi$-parameterized seed model $\varepsilon_{\phi_0}$—a dense, decoder-only transformer pretrained on chess language modeling.
2. **Train.** Each model copy $\varepsilon_{\phi_e}$ undergoes independent, asynchronous fine-tuning on game transcripts featuring a designated GM playing as either White or Black. This phase produces "persona" models that have refined their move distribution toward the stylistic tendencies of their reference player. We refer to these models $\varepsilon_{\phi_e}$ as experts.
3. **Stitch.** MoM is assembled from experts ($\varepsilon_{\phi_e}, e > 0$) using a hybrid approach. At each layer, we either apply a weight-merging algorithm or implement a router to gate access to the original expert weights. Training is confined to the routing modules and the newly-formed merged-weight layers, which undergo an alignment phase.

---

[1]We use the term *"move"* to refer to an individual action by either White or Black (i.e., a semi-move or ply). We use the term *"turn"* for a complete move cycle, commonly known as a full move.

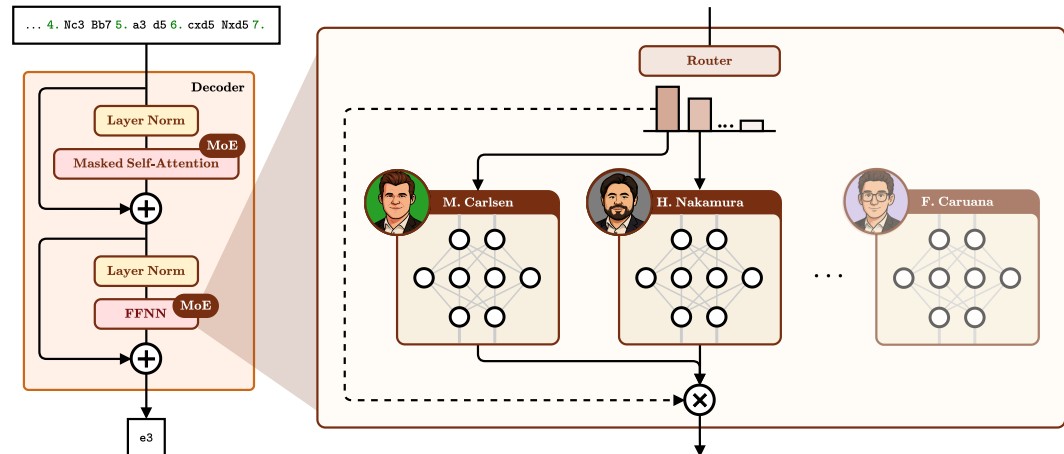

Figure 1: **Illustration of MoM.** First, multiple decoder-only chess language models are trained to emulate the game decisions of specific grandmasters. Then, their layers are combined into a sparse language model by alternating uniform weight merging and top-$k$ routing for next move prediction.

In line with standard practice for chess language modeling, MoM operates on games in Portable Game Notation (PGN) using a 32-character input-output vocabulary (see Appendix C.2), which ensures broad compatibility with existing seed models.

## 3.1 GRANDMASTER EXPERTS

Each expert model $\varepsilon_{\phi_e}$ is derived through a two-phase fine-tuning process, first to capture a GM fingerprint (i.e., approximate a player's move distribution) and then to enforce game-rule adherence.

**SSL.** We train $\varepsilon_{\phi_e}$ to autoregressively predict the moves of its target $p$-th player. This is achieved by computing the cross-entropy exclusively on the considered player's tokens, excluding opponent ones. Let $\mathcal{D}_e = \{(s, m)\}$ denote the state–move pairs from the expert dataset, where $m$ is executed by $p$ starting from the game state $s$. Our *player-side loss* is defined as: $\mathcal{L}_{\text{SSL}} = -\sum_{(s,m) \in \mathcal{D}_e} \log \varepsilon_{\phi_e}(m|s)$.

**RL.** SSL alone may produce experts that generate suboptimal or illegal moves due to overfitting or distributional shift. For this reason, we further refine $\varepsilon_{\phi,e}$ using the Group Relative Policy Optimization (GRPO) algorithm (Shao et al., 2024). Let $s$ denote the board state of the considered game. We sample a set of $M$ candidate next moves $\{m_i\}_{i=1}^{M} \sim \varepsilon_{\phi_e}(s)$ via temperature-controlled decoding. Each move candidate is evaluated along two axes: (1) *syntactic correctness* $\rho_{\text{synt}}$, whether the PGN substring is well-formed; (2) *legality* $\rho_{\text{leg}}$, whether the move conforms to chess rules. The computed reward values $\mathbf{r} = \{r_1, \ldots, r_M\}$ serve as a guiding signal to promote correct actions. More formally, the policy optimization objective is expressed as:

$$\mathcal{J}_{\text{GRPO}} = \frac{1}{M} \sum_{i=1}^{M} \min\left(\mathcal{R}_i \cdot \hat{A}_i; \text{clip}_{1\pm\epsilon}(\mathcal{R}_i) \cdot \hat{A}_i\right) - \beta \, \mathbb{D}_{KL}\left(\varepsilon_{\phi_e} \| \varepsilon_{\phi_e^{\text{old}}}\right)$$

$$\text{with} \ \ \mathcal{R}_i = \frac{\varepsilon_{\phi_e}(m_i|s)}{\varepsilon_{\phi_e^{\text{old}}}(m_i|s)}; \ \hat{A}_i = \frac{\rho_{\text{synt}}(m_i) + \rho_{\text{leg}}(m_i) - \mu_{\mathbf{r}}}{\sigma_{\mathbf{r}}}$$

(1)

where the advantage $\hat{A}_i$ is normalized across the batch of candidate moves. All tokens forming a single candidate move inherit the same cumulative reward. By adding an RL training stage, we not only incentivize move exploration, but also tackle the "memorization syndrome"—which we posit is the main barrier preventing chess language models from rivaling engines empowered with external search. "*Education in chess has to be an education in independent thinking and judging. Chess must not be memorized, simply because it is not important enough.*"– E. Lasker; World Chess Champion 1894-1921.

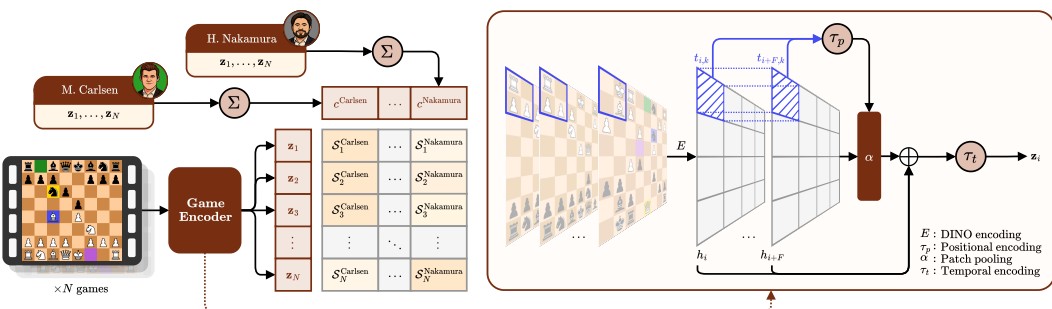

Figure 2: **Overview of the visual chess player identification system.** *Left:* During training, game embeddings are processed through contrastive learning against GM-specific centroids to enforce intra-player similarity and inter-player distinctiveness. *Right:* The visual encoding pipeline processes consecutive chess board frames to extract and temporally aggregate spatial patch tokens (in blue), with positional and temporal encodings generating the final game embedding.

## 3.2 STITCHING

MoM employs a hybrid parameter composition strategy: $\Phi_{\text{MoM}} = \Phi_{\text{gated}} \cup \Phi_{\text{shared}}$.

$\Phi_{\text{gated}}$ comprises expert-specific layers subjected to dynamic routing. From the decoder blocks of each expert, we retain and parallelize the linear layers before and after masked self-attention, specifically the $Q$-$K$-$V$ and output projection layers. Before each parallelization point, a learnable linear gating network $\mathcal{G}_\phi$ is introduced to mobilize the appropriate transformations, mapping the current board state $s$ to a probability distribution over experts: $P(e|s) = \text{softmax}(\mathcal{G}_\phi(s))$. During inference, only the top-$k$ experts with the highest routing probabilities are activated, using a weighted sum pooling: $\sum_{e \in \text{top-}k(P(e|s))} P(e|s) \cdot \varepsilon_{\phi_e}(s)$.

To enhance differentiable top-$k$ selection during training while preventing mode collapse, we employ Gumbel-Softmax with temperature annealing (Jang et al., 2017). The temperature is gradually decreased during training to transition from exploration to exploitation, naturally enforcing load balancing by encouraging diverse expert selection in early phases while eventually converging to sharp, efficient routing (Fedus et al., 2022).

All remaining parameters not subject to gating—including token embeddings, attention heads, and extra FFNNs—are merged across experts using uniform averaging to create a shared backbone $\Phi_{\text{shared}}$. Further information about merging techniques and their effects on downstream performance are reported in Appendix D.

## 3.3 BEHAVIORAL STYLOMETRY

Chess players exhibit subtle stylistic preferences that manifest as tactical tendencies in specific game scenarios. Unlike average players, GMs possess comprehensive knowledge across all phases of the game and excel across diverse strategic motifs. Traditionally these traits have been described qualitatively by human analysts. However, the widespread adoption of AI-powered engines for preparation and analysis has made such distinctions increasingly difficult to substantiate. To this end, we develop a stylometry framework that embeds games into a representation space capturing both spatial board structure and temporal evolution.

Let a game $g$ by the $p$-th player be represented as a sequence of video frames $\mathcal{V}_g^p = \{I_1^p, I_2^p, \ldots, I_T^p\}$, where $I_j^p$ denotes the board configuration following the $j$-th move by $p$. We extract fixed-length subsequences $\mathcal{F}_g^p$ of size $F$ to standardize input sequences and capture stylistic traits at different stages of play. A pretrained vision transformer $E_\psi$ processes each frame $I_j^p$ to extract $L$ patch-token embeddings $\{t_{j,k}^p\}_{k=1}^L$ (Figure 2). We aggregate embeddings along two axes: temporally across the frame window $[i, i + F - 1]$, where each sub-area $k$ receives $r_k^p = \frac{1}{F} \sum_{j=i}^{i+F-1} t_{j,k}^p$; and spatially for each frame $j$, yielding $h_j^p = \frac{1}{L} \sum_{k=1}^L t_{j,k}^p$. These two views are fused to produce frame-level representations that capture both local region evolution and global board state, where $\alpha$ is a learned attention-based transformation over the temporally-smoothed patch features. To encode sequential

dependencies across distinct phases of the game, we further augment each frame representation with positional embeddings $\tau_p(j)$ and process the resulting sequence via a LSTM network $\tau_t$. The final game embedding, denoted $z_g^p \in \mathbb{R}^d$, is defined as:

$$\mathbf{e}_j^p = h_j^p + \alpha\left(\{r_k^p\}_{k=1}^L + \tau_p(j)\right), \quad \mathbf{z}_g^p = \tau_t(\{\mathbf{e}_i^p, \mathbf{e}_{i+1}^p, \ldots, \mathbf{e}_{i+F-1}^p\}). \tag{2}$$

Drawing inspiration from speaker recognition systems and McIlroy-Young et al. (2021), we employ the generalized end-to-end (GE2E) loss (Wan et al., 2018) to map $\mathcal{V}_g^p$ representations into clusters of GMs' games. Let $N$ denote the number of players in a batch and $M = |G_p|$ the number of games per player. For all players $p,q$ and game $g$, pairwise similarity scores $\mathcal{S}_g^{p,q}$ are computed between each game embedding $\mathbf{z}_g^p$ and player-specific centroids $c_g^p$, accounting for within-player game $g$ contamination in centroid computation:

$$\mathcal{S}_g^{p,q} = W \cdot \cos\left(\mathbf{z}_g^p, c_g^q\right) + b, \quad \text{where} \quad c_g^q = \begin{cases} \frac{1}{M-1} \sum_{\tilde{g} \in G_q \setminus \{g\}} \mathbf{z}_{\tilde{g}}^q & \text{if } p = q \\ c^q := \frac{1}{M} \sum_{\tilde{g} \in G_q} \mathbf{z}_{\tilde{g}}^q & \text{if } p \neq q \end{cases} \tag{3}$$

where $W, b \in \mathbb{R}$ are learnable scaling parameters. The training objective follows InfoNCE (van den Oord et al., 2018) with additional regularization terms:

$$\mathcal{L}_{\text{style}} = -\frac{1}{NM} \sum_{p=1}^N \sum_{g \in G_p} \log \frac{\exp(\mathcal{S}_g^{p,p})}{\sum_{q=1}^N \exp(\mathcal{S}_g^{p,q})}$$

$$+ \frac{\lambda_m}{N(N-1)} \sum_{\substack{p,q=1 \\ p \neq q}}^N \max(0, \cos(c^p, c^q) + \mu) + \frac{\lambda_c}{NM} \sum_{p=1}^N \sum_{g \in G_p} \left(1 - \cos(\mathbf{z}_g^p, c_g^p)\right) \tag{4}$$

where $\lambda_m$ and $\lambda_c$ are regularization weights, and $\mu$ is the margin parameter. The margin loss enforces inter-player separation, while the centroid loss promotes intra-player compactness by minimizing cosine distances between embeddings and their centroids. This formulation aligns embeddings with their reference centroid while keeping them separable from others, inducing stylistic coherence within individuals and distinctiveness across the population of GMs.

## 4 EXPERIMENTS

### 4.1 EXPERIMENTAL SETUP

**Datasets.** Experts We anchor our exploratory research on 10 GMs, selected for their high coverage in chess databases. We systematically acquire their game records from three sources: PGN-Mentor (64 Squares, 2025), a chess archive with >1M GM games primarily from over-the-board tournaments; Chess.com (Chess.com, LLC, 2007) and Lichess (Thibault Duplessis et al., 2010), the largest online chess platforms. For each GM, the collected games are sampled to ensure a balanced representation of the instances played as White and Black; we allocate 80% for training and 20% for testing. Gating Following Zhang et al. (2025a), upon initialization, the router components within the MoM models are trained on a data mixture that includes 50% of the pretraining games from the seed model, the other half consisting of equal proportions of games from the training set of each mounted GM. Behavioral stylometry For the player classification task–in contrast to player emulation–we create a dataset where each GM is represented by an equal number of training games. We set a uniform size of 1,000 games, a data threshold that all GMs in our cohort meet. Stratified sampling is applied to form individual sets that preserve a balanced White-Black color distribution. Each PGN string is transformed into a video for our vision-based models. This sequence is constructed by generating a frame for each move played by a target GM, while the opponent's replies are disregarded. To make the action within each frame explicit, the move's starting and ending tiles are highlighted with color. The board's perspective is standardized across all frames: it is always oriented so that the target GM's pieces are at the bottom, which involves rotating the board when they are playing as Black. All We restrict our interest to games classified under the Blitz and Rapid categories, where players have between 3 and 30 minutes per game. We exclude faster formats, as these involve a higher frequency of mistakes, sometimes intentional. We filter out games shorter than 5 moves, as these are often pre-arranged draws or database errors. Duplicate games are resolved by

Table 1: **Grandmaster dataset statistics.** Aggregated view (train, test). Played games span from 1984 to 2025.

| Grandmaster | Elo (Avg±Std) | # Games[†] | 🏆 (%)[‡] | # Moves / Game | | |
|---|---|---|---|---|---|---|
| | | | | Min | Avg | Max |
| ❶ V. Anand | 2,752 ± 16 | 4,475 | 34 | 11 | 81 | 290 |
| ❷ L. Aronian | 2,794 ± 97 | 6,452 | 37 | 10 | 92 | 321 |
| ❸ M. Carlsen | 2,940 ± 161 | 8,466 | 49 | 9 | 93 | 348 |
| ❹ F. Caruana | 2,799 ± 51 | 6,658 | 45 | 6 | 98 | 340 |
| ❺ A. Firouzja | 2,792 ± 107 | 5,114 | 50 | 8 | 94 | 359 |
| ❻ A. Giri | 2,746 ± 43 | 6,886 | 37 | 10 | 92 | 349 |
| ❼ H. Nakamura | 2,902 ± 184 | 10,016 | 51 | 7 | 92 | 400 |
| ❽ I. Nepomniachtchi | 2,792 ± 72 | 13,238 | 50 | 6 | 92 | 323 |
| ❾ W. So | 2,819 ± 117 | 11,764 | 48 | 6 | 88 | 396 |
| ❿ M. Vachier-Lagrave | 2,821 ± 133 | 1,220 | 37 | 10 | 93 | 253 |

[†] 37,002 from PGN Mentor, 28,243 from Chess.com, 9,044 from Lichess.
[‡] Proportion of games won.

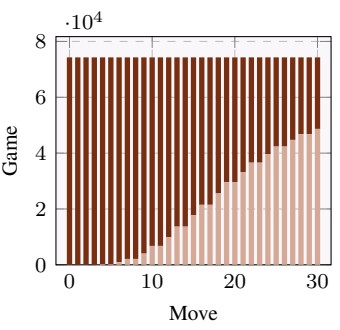

Figure 3: **Distribution of unique games (●) by move.**

discarding the Chess.com copy. Table 1 provides an aggregated view of dataset composition before splitting. The average Elo rating of the players, recorded at game time, is 2,816. Figure 3 illustrates that, beyond move 15, fewer than 25% of the games played by the GMs remain unique. Refer to Appendix C.1 for a complete breakdown on dataset construction and statistics.

**Models.** Seed We consider four autoregressive transformer decoders, each grounded in 50M-parameter nanoGPT models trained from scratch on millions of Lichess records (Karpathy, 2022). The first three models are from the collection released by (Zhang et al., 2024). Each of these models, denoted $T_t$, was trained exclusively on games by players rated below a specific Glicko-2 threshold, $t$, but eventually transcended to a higher rating, $t'$, at test time. More precisely, our experiments involve: $T_{1,000}$ ($t' \approx 1,500$), $T_{1,300}$ ($t' \approx 1,500$), $T_{1,500}$ ($t' \approx 1,500$). The fourth model, developed by Karvonen (Karvonen, 2024), represents his largest implementation and has been trained without imposing rating restrictions. Experts We hypothesize that the optimal skill-level checkpoint for style acquisition may vary depending on the GM. To test this, each seed model is fine-tuned on the games of a distinct GM. For SSL, we train over 6K steps using a batch size of 8 and a learning rate of 2e-6. For RL, we train over 6K steps with 8 groups of $M = 8$ candidate next moves for batch, and a learning rate of 6e-7. The reward function is a combination of $\rho_{\text{synt}}$ for correct format signal, and $\rho_{\text{leg}}$ for proximity to the closest legal move (computed with edit distance). MoE We stitch only the five experts with superior performance in evaluation metrics. This choice is predicated on maximizing the ensemble's efficacy while maintaining a lightweight and tractable scope for our experimental analysis. Behavioral stylometry We opt for DINOv3 (Siméoni et al., 2025) with 21.6M parameters as $E_\psi$, a SOTA self-supervised vision transformer for image encoding. $E_\psi$ was pre-trained for 15k steps on a classification task to incentivize attention toward the board area of the next move. This initializes embeddings to capture the nuanced features of a specific chess position. We fine-tuned it for 25K steps with in-batch negative samples composed of $N = 10$ and $M = 5$, with the number of frames $F$ set to 5. The opening phase is discarded from the training data due to the low diversity among GMs (see Appendix C.1). Reproducibility is ensured by fixing the random seed to 960–the "*Fischer seed*". See Appendix C.2 for details on implementation and hyperparameters.

**Metrics.** Stockfish battle Average win and draw rates achieved by the model when playing 100 games against Stockfish 16.1 at a specified level, repeated 10 times for statistical robustness. Each level directly controls Stockfish's skill setting, search depth, and time limit–the higher the level, the stronger the opponent. We constrain Stockfish to evaluate up to 100K nodes per move without a time cap; this operational mode substantially reduces computational requirements while eliminating inconsistencies that might arise from hardware variations or processing load fluctuations (Karvonen, 2024). The chess language model under evaluation uses a greedy decoding strategy, while Stockfish's moves are randomized by applying a temperature of 1 to the probability distribution derived from centipawn evaluations. The game proceeds in a turn-based manner: after each move by the model, the updated board state is passed to Stockfish, and vice versa. The model operates under a strict no-retry policy; the generation of a single illegal move results in an immediate forfeiture of the game. The selected seed models, along with our derivative models, have a maximum input length of 1,023 tokens that accommodate ∼92 turns (184 moves) in PGN format. Consistent with Karvonen (Karvonen, 2024), games are forcibly ended after 90 turns and the outcomes are resolved

by centipawn evaluation of the final board state.[2] In each match-up, the model and Stockfish swap seats to ensure fair White and Black opening exposure. To better align with traditional tournament evaluation, we also calculate an aggregate score we term FIDEScore, awarding 1 point for a win, 0.5 for a draw, and 0 for a loss. Legality Percentage of games not ended because of an illegal move generated by the model when playing against Stockfish 16.1 under the previously described settings.

**Baselines.** We evaluate the experts both in isolation and as part of the MoM ensemble. In addition, we compare MoM with model soup (Wortsman et al., 2022), where expert and seed model weights are uniformly averaged without further fine-tuning. We also benchmark against the seed models. As emphasized in prior work (Ruoss et al., 2024), we caution that a direct and fair comparison with other engines comes with significant caveats, as they employ FEN input representations, follow different training protocols, and may utilize search at test time. We situate MoM models within the broader landscape in Appendix B. However, we note that some conclusions can only be drawn within our family of models and the corresponding ablations that keep all other factors fixed.

## 4.2 RESULTS

We organize our experimental results around a series of research questions (RQ1-RQ6) that interrogate different dimensions of our approach, from the impact of design decisions to MoM capabilities.

**RQ1** **Which seed model is the optimal foundation for training grandmaster experts?** The findings in (Zhang et al., 2024) demonstrate that there is no clear correlation between a model's initial capabilities and its achievable performance after fine-tuning, and that even a higher WinRate may result in worse move legality because of knowledge forgetting. Consequently, identifying the optimal seed model is a non-trivial task, as the fine-tuning process can significantly influence the initial model choice. This implies that the selection cannot be reduced to simply choosing the highest-performing base model. To empirically determine the most effective foundation for our grandmaster experts, we report in Figure 4 the comparative performance after fine-tuning from several compatible seed models (Karvonen, 2024; Zhang et al., 2024).

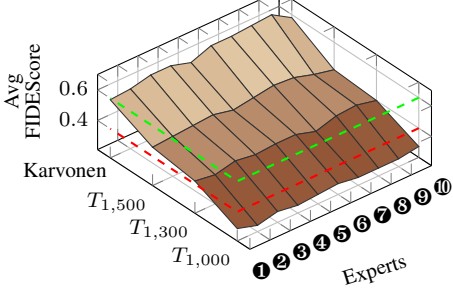

Figure 4: **Effect of seed model on expert FIDEScore.** SSL-only. Stockfish 1, pooled over 10 runs.

**RQ2** **Does SSL + RL result in greater legality than SSL alone?** Our results (Figure 5) show that incorporating RL into next-move prediction reduce the illegality rate in chess language models. As noted in prior work (Zhang et al., 2025a), while SSL-trained models can exhibit creative and advanced play, they can suffer in terms of move legality—likely due to forgetting pretraining knowledge. This issue may stem from the distributional shift in game states that individual experts encounter. To mitigate such inconsistencies, our MoE architecture is essential, enabling robustness by balancing the strengths and limitations of each specialized expert. The higher legality metric demonstrates how reinforcement learning steers chess language models toward accurate move selection and deeper understanding of board positions, enabling comprehension of strategic implications and more sophisticated, grandmaster-level decision-making processes.

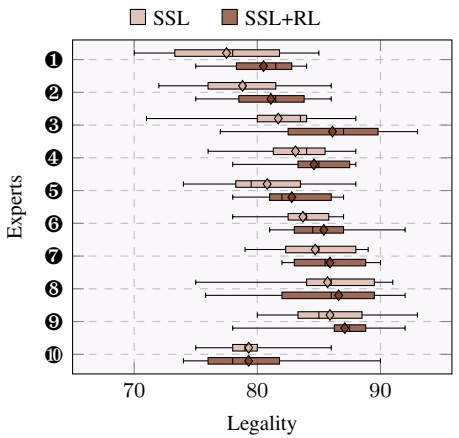

Figure 5: **Effect of RL on legality.** Karvonen seed. Stockfish 1, pooled over 10 runs.

---

[2]Stockfish provides an evaluation in centipawns, which we convert to a win probability with the formula: Win% $= 100/(1+\exp(-0.00368208 \times \text{centipawns}))$ from https://lichess.org/page/accuracy.

Table 2: **Effect of RL on game results.** Karvonen seed. Stockfish 1, pooled over 10 runs. The top-5 experts, balancing legality and FIDEScore, are bolded.

| | Metric[†] | ❶ | ❷ | ❸ | ❹ | ❺ | ❻ | ❼ | ❽ | ❾ | ❿ |
|---|---|---|---|---|---|---|---|---|---|---|---|
| SSL | Draw Rate | $14.7_{\pm3.0}$ | $14.6_{\pm3.7}$ | $15.2_{\pm3.7}$ | $14.4_{\pm3.1}$ | $15.8_{\pm5.3}$ | $16.2_{\pm3.1}$ | $16.1_{\pm2.6}$ | $13.5_{\pm4.3}$ | $18.5_{\pm4.9}$ | $16.3_{\pm1.8}$ |
| | Win Rate | $52.0_{\pm4.5}$ | $52.6_{\pm4.2}$ | $55.0_{\pm5.4}$ | $55.3_{\pm4.1}$ | $51.2_{\pm6.5}$ | $55.6_{\pm5.2}$ | $55.4_{\pm3.1}$ | $58.5_{\pm5.5}$ | $56.4_{\pm5.0}$ | $49.6_{\pm4.1}$ |
| | FIDEScore | $59.4_{\pm3.8}$ | $59.9_{\pm4.6}$ | $62.6_{\pm5.3}$ | $62.5_{\pm4.6}$ | $59.1_{\pm5.5}$ | $63.7_{\pm4.4}$ | $63.5_{\pm4.0}$ | $65.3_{\pm4.1}$ | $65.6_{\pm4.4}$ | $57.8_{\pm4.1}$ |
| SSL +RL | Draw Rate | $15.8_{\pm4.4}$↑ | $20.5_{\pm2.5}$↑ | $23.3_{\pm4.7}$↑ | $20.4_{\pm3.8}$↑ | $18.7_{\pm3.0}$↑ | $19.4_{\pm4.1}$↑ | $20.9_{\pm3.9}$↑ | $19.4_{\pm5.8}$↑ | $22.6_{\pm5.7}$↑ | $17.1_{\pm3.7}$↑ |
| | Win Rate | $51.4_{\pm4.9}$↓ | $47.3_{\pm3.1}$↓ | $51.1_{\pm4.3}$↓ | $48.5_{\pm4.7}$↓ | $51.6_{\pm5.0}$↑ | $54.1_{\pm4.4}$↓ | $53.8_{\pm4.4}$↓ | $53.2_{\pm5.3}$↓ | $52.7_{\pm4.5}$↓ | $49.8_{\pm3.9}$↑ |
| | FIDEScore | $59.3_{\pm4.0}$↓ | $57.6_{\pm2.6}$↓ | **$62.8_{\pm3.8}$↑** | $58.7_{\pm3.5}$↓ | $61.0_{\pm4.2}$↑ | **$63.8_{\pm3.3}$↑** | **$64.3_{\pm3.1}$↑** | **$62.9_{\pm3.3}$↓** | **$64.0_{\pm3.4}$↓** | $58.4_{\pm4.9}$↑ |

† All reported values are Avg±Std (%). Seed model: 24.0±2.4 (Draw Rate), 42.1±4.0 (Win Rate), 54.1±4.1 (FIDEScore).

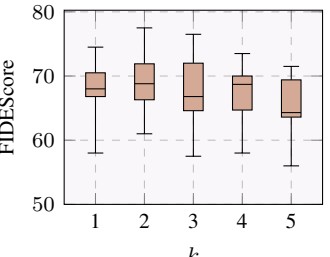

Figure 6: **Style Consistency** (left): Relative change in cosine distance when computing expert-specific centroids from random subsamples of played games; **Style Acquisition** (right): Recall of style-similarity retrieval mapping of played games to the correct real-GM centroid.

**RQ3** **How does RL affect playing style?** While RL training improves most expert models in FIDEScore (Table 2), WinRate slightly decreases in some configurations. The higher DrawRate across all setups, along with qualitative game analysis, indicates RL models adopt a more cautious style. Although mid-game accuracy improves, RL models often fail to execute the final checkmate, leading to more draws–even from net winning positions–whereas SSL models pursue riskier lines regardless of potential illegal moves. We argue that a slight WinRate decrease is less critical than maintaining consistent legal play.

**RQ4** **Can MOM outperform dense models?** Figure 8 demonstrates that MOM consistently outperforms all baseline approaches across Stockfish difficulty levels 0–5. While individual expert models excel in capturing specific GM playing styles, they often underperform in diverse strategic situations. MOM emerges as the most well-rounded generalist, dynamically accessing appropriate expert knowledge through learned routing mechanisms, and outperforms the model soup baseline (Wortsman et al., 2022) by up to +3 FIDEScore, demonstrating that intelligent gating networks surpass naive parameter averaging. This performance advantage is maintained even as game difficulty increases, with MOM showing more graceful degradation than all baselines.

Figure 7: **Effect of activated expert count** $k$ **on game results.** MOM (top-5 exp. by FIDEScore), Stockfish 0, pooled over 10 runs.

**RQ5** **Can expert models acquire GMs' stylistic traits?**
Identifying elite players' stylistic patterns is challenging, as classification-based methods achieve limited accuracy (McIlroy-Young et al., 2021). Within our retrieval-based framework, Figure 6 reports the normalized cosine similarity between centroid embeddings of actual grandmaster and expert-generated games. To evaluate *style consistency*, we partition each expert's games collection, compute a centroid on one subset, and assess similarity with the complementary set at different ratios. The small relative drift across splits indicates stable, self-consistent stylistic behavior. For *style acquisition*, nearest-centroid retrieval against real GM embeddings shows each expert reliably ranks its designated master among the closest matches. While retrieval accuracy is marginally higher when evaluated on real grandmaster games, performance on expert-generated games remains comparable, demonstrating that experts operating well below $2,600$ Elo successfully reproduce distinctive and identifiable master-specific stylistic signatures.

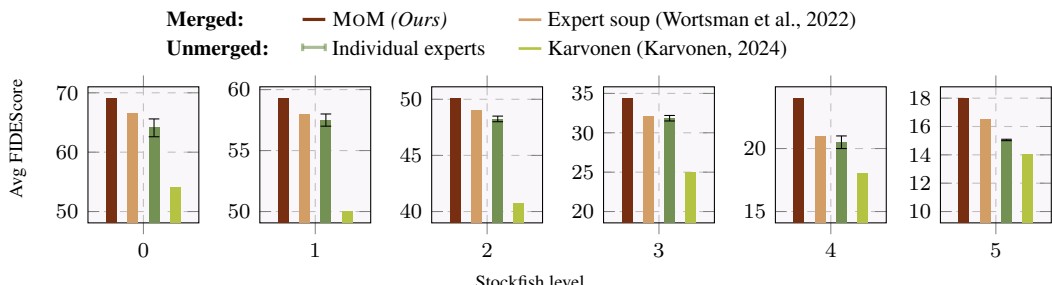

Figure 8: **Comparison between MOM (top-5 experts by FIDEScore, SSL+RL, Karvonen seed) and baselines.** FIDEScore after battling Stockfish at increasing difficulties; average results after 10 runs for each level.

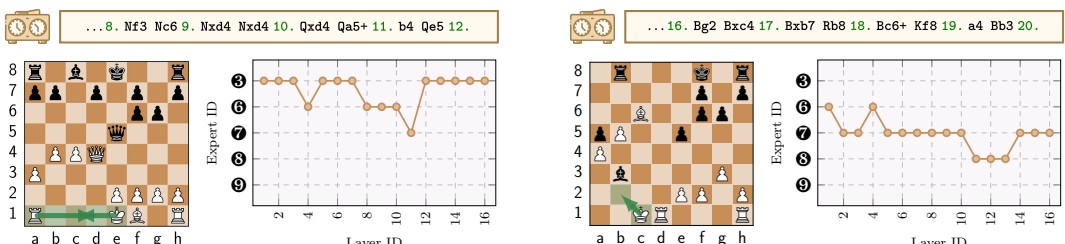

Figure 9: **Visualization of how MOM activated experts vary when playing a game at test time against Stockfish.** Decoder block top-1 routing paths for two distinct board states. MOM (White) dynamically adjusts expert utilization in response to the evolving position.

**RQ6** **Do MOM activation patterns reflect interpretable and meaningful style transitions?** The MoE gating weights determine the influence of each expert on the combined representation used to predict the next move. To probe MOM's decision-making, we analyze the top-1 activated expert in each decoder block during a game against Stockfish (Figure 9). Our final model employs top-$k$ experts per token, using $k$ of 2 based on empirical evaluation (Figure 7), as this balances expert diversity with routing precision, while larger values progressively degrade performance by introducing noise that dilutes expert specialization. Expert activations align with the distinctive strengths of individual players, effectively complementing one another and shifting over time. In the example, MOM's play changes clearly. In the early midgame, it adopts a fearless, aggressive posture reminiscent of a young Magnus Carlsen—castling queenside with `12.O-O-O` despite Black's threats. Later, MOM takes a more imaginative and tactical style. On move `20.Kb2`, it sacrifices a rook to fuel the attack—an idea strongly evocative of Nakamura.

## 5 CONCLUSION

This paper challenges the conventional procedure of training dense chess language models on aggregated, player-undistinguished datasets. We introduce MOM 5×50M, the first chess MoE to combine independently trained GM networks into a stronger and controllable model—demonstrating improved performance statistics in games against Stockfish over individual experts and preventing stylistic homogenization. We create GM networks in two stages, pairing player-centric next move prediction with GRPO for improved legality. The sparse model is built by alternating weight merging and lightweight player routing mechanisms. Moreover, we train an image encoder to classify a GM by a sample of its moves, making it possible to verify expert specialization and interpret MOM behavior. Our experimental results demonstrate that individual expert models achieve significantly improved performance through our SSL+RL training recipe, and crucially, that the sparse MoE architecture consistently outperforms these individual experts in games against Stockfish. These findings validate both our persona specialization approach and the effectiveness of MoE architectures for chess language modeling, establishing that compositional AI systems can successfully preserve stylistic diversity while achieving superior performance. We follow rigorous open science principles. Collectively, our contributions set the groundwork for a new generation of decentralized and compositional chess AI.

## ETHICS STATEMENT

Our introduction of a behavioral stylometry metric, aimed at verifying the distinctiveness of AI personas, necessitates a careful consideration of its ethical implications. Although our primary motivation is to advance human-compatible AI by creating chess language models with recognizable styles, we acknowledge that stylometry, in general, can be used in ways that compromise individual privacy. A sufficiently accurate stylometry model could potentially be used to deanonymize players across different platforms or accounts, potentially revealing ideas they are experimenting with or details about when they are active online. This is particularly concerning for engaged players who wish to play without connecting that activity to their public persona. Our research confirms that this vulnerability exists even at the highest competitive level, affecting GMs rated above 2,700 Elo. Notably, this result was achieved using a model trained on what is, to our knowledge, the smallest corpus of games yet reported for such a task. The history of author identification makes clear that people have an interest in developing countermeasures, and our work offers new insights for those seeking to protect their anonymity in chess. Although prior research identified the opening as the most stylistically revealing phase for intermediate players, our findings for the GM bracket show the opposite. At this elite level, the most frequent opening lines are so homogeneous that they provide a weak signal for identification. To effectively obscure their identity, a player must instead alter the characteristic patterns that emerge in the less-theorized middlegame and endgame phases. These include both strategic preferences (e.g., the handling of specific pawn structures or piece imbalances) and technical tendencies (e.g., executing complex endgame conversions or establishing defensive fortresses). Although any such countermeasure requires conscious effort, our contributions enable a more targeted approach. Furthermore, by analyzing attention weights on visual patches, our framework offers a promising pathway toward designing subtle safeguards that require minimal individual adaptation. We acknowledge that the learned embeddings could inadvertently capture demographic attributes, such as nationality, by associating them with playing styles. Investigating this possibility was beyond the scope of our work; however, it represents a known risk of unintended bias in deep learning. Because our method operates on raw visual inputs (i.e., frame sequences), it obviates the need for domain-specific feature engineering, rendering it broadly applicable to sequential decision-making tasks in domains as diverse as video games and medicine. This portability highlights the importance of proactively establishing ethical guidelines and quantifying privacy risks before similar techniques are deployed in higher-stakes environments. We hope that the research community continues to develop the methodological and ethical frameworks necessary to ensure that behavioral stylometry techniques improve human-AI compatibility and collaboration.

## REPRODUCIBILITY STATEMENT

In accordance with open-science principles, data, code, and model weights are publicly available at `https://anonymous.4open.science/r/mixture-of-masters` (MIT License). Consistent with ethical considerations, the weights for behavioral stylometry classification are not publicly released. We provide a Reproducibility Appendix (C) to fully document our methodology. This includes a description of the data sources, filtering criteria, and pre-processing pipelines (C.1), as well as details on implementation and experimental setup, including all model hyperparameters, training configurations, and the computational infrastructure used (C.2).

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

## A  SURVEY

In parallel to the methodological and resource contributions presented in the main paper, we designed and administered a survey aimed at clarifying long-standing open questions in the chess community that directly underpin our modeling approach. In particular, the survey seeks to explore the viewpoint of participants on four fundamental dimensions: (i) the perceived possibility of identifying professional players through the sole observation of their games, (ii) the existence and definition of the notion of "playing style," (iii) the practical feasibility of assigning coherent style categories to GMs, and (iv) the extent to which chess engines and AI models influence human play. Since MOM presupposes assumptions about style and player recognition—concepts that remain contested even among experts—this empirical complement serves as a critical validation step.

We selected the *Alma Mater Studiorum Chess Tournament 2025*[3] as primary venue for data collection. An international academic competition organized by the University of Bologna and held behind closed doors from September 12 to 14, 2025, at the Biblioteca Universitaria di Bologna, Italy. The event convened 72 mixed-gender players, grouped into 18 teams of four members each, representing some of the world's most prestigious universities from 10 countries across three continents. The selection process for these teams was notably rigorous, as each institution was responsible for fielding its most talented representatives, often through internal qualification tournaments. Consequently, the participant pool—composed entirely of adult English-speaking students (from bachelor to Ph.D. level) and faculty members—included players of exceptional caliber, among them national champions. The tournament structure consisted of five playing sessions governed by the Swiss system with a time control of 45 minutes plus a 10-second increment per move. It was overseen by arbiters from the Italian Chess Federation and was not rated by FIDE to preserve its inclusive and collegial character, prioritizing cultural exchange and sportsmanship. The event received live commentary on Chess.com channels[4] and featured an AI analysis room sponsored by Intel.

The decision to anchor our study in this specific tournament was deliberate to ensure the collection of high-quality and reliable data from a culturally diverse participant base. In stark contrast to large-scale online surveys, where participant veracity and expertise can be difficult to ascertain, this setting provided a controlled environment with a verified cohort of competent players. The context also provided an atmosphere of intellectual openness and reflection, well suited for a survey.

**Data collection.**  Our data collection protocol was executed in two distinct phases. The first phase took place in person during the three days of the tournament. This direct interaction encouraged thoughtful, authentic responses, collected in an environment free from external distractions. The closed-door format of the event allowed us to engage not only players but also arbiters and AI experts, thereby broadening the scope of informed perspectives. Recognizing that the demanding tournament schedule could limit participation, we initiated a second phase post-event. An online version of the survey was made available for a limited period to allow contributions from individuals who were unable to complete it on-site, as well as to include additional voices from the broader chess community, such as members of chess clubs who did not attend the tournament. Throughout both data collection phases, strict ethical and procedural standards were maintained. We ensured all respondents were over the age of 18 and obtained their informed consent. The submission of responses was strictly voluntary, without financial or other incentives. Survey users were not shown their previous answers and aggregate results during or after the collection process, a measure implemented to mitigate potential conformity biases. On average, completing the survey required approximately eight minutes. To guarantee participant privacy, the survey was designed to be fully anonymous. No personally identifiable information—such as names, email addresses, or IP addresses—nor any other sensitive data was gathered.

### A.1  PARTICIPANT GEOGRAPHY AND DEMOGRAPHICS

The survey solicited non-identifying, high-level demographic information: affiliation name, affiliation country, and current Elo rating. For each of these items, a *"Prefer not to say"* option was supplied to respect respondent privacy. Our sample covers a broad heterogeneity in both demographic and geographic terms, with 50 responses obtained from all the 18 competing universities,

---

[3]https://events.unibo.it/alma-mater-university-chess-tournament
[4]https://www.chess.com/it/events/alma-mater-chess-tournament-2025

arbiters and independent experts. This diversity was crucial, allowing us to capture opinions across multiple chess traditions and educational backgrounds. Simultaneously, the shared academic context delivered sufficient common ground to ensure meaningful comparability of responses. Our respondents included members of teams from Yale and Harvard in the United States, and from Oxford and Cambridge in the United Kingdom—pairs of institutions whose chess rivalries trace back more than 150 years. Geographic breakdowns are presented in Figure 10, while demographic characteristics are summarized in Table 3.

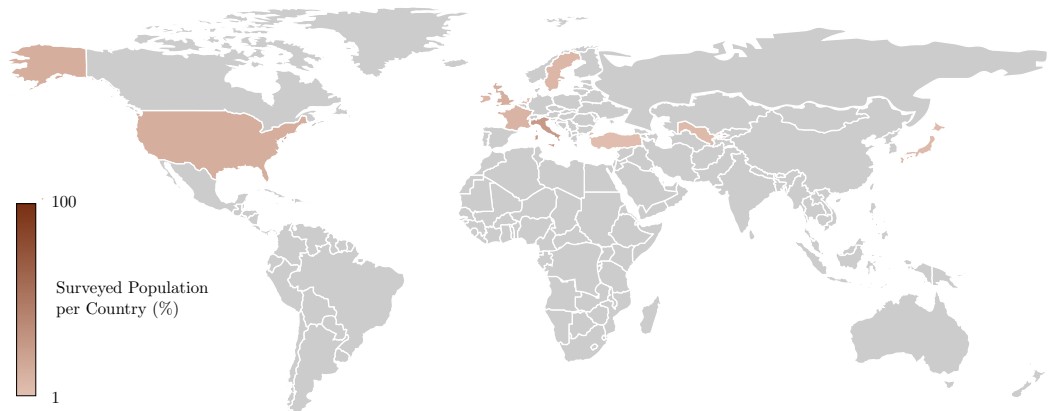

Figure 10: **Geographic distribution of survey participants by affiliation country.**

Table 3: **Demographics of survey participants** ($N = 50$). Overall distribution of Elo ratings. Counts and percentages of participants by affiliation.

| Elo | Continent | Country | Affiliation[†] | Participants | |
|---|---|---|---|---|---|
| | | | Alma Mater Studiorum – Università di Bologna | 4 | 8 % |
| | | | Università di Pisa | 2 | 4 % |
| | | Italy | Università degli Studi di Padova | 2 | 4 % |
| | | | Università degli Studi di Milano Bicocca | 1 | 2 % |
| | | | Università degli Studi di Napoli Federico II | 1 | 2 % |
| | | | Other | 3 | 6 % |
| | | | University of Oxford | 2 | 4 % |
| | | United Kingdom | University of Cambridge | 2 | 4 % |
| | | | Other | 2 | 4 % |
| | Europe | | Trinity College Dublin | 2 | 4 % |
| | | Ireland | University College Dublin | 2 | 4 % |
| | | | Other | 1 | 2 % |
| | | France | Université Paris 1 Panthéon Sorbonne | 2 | 4 % |
| | | | Other | 2 | 4 % |
| | | Netherlands | Maastricht University | 2 | 4 % |
| | | | Eindhoven University of Technology | 1 | 2 % |
| | | Sweden | Lund University | 2 | 4 % |
| | | | Other | 1 | 2 % |
| | North America | United States of America | Harvard University | 2 | 4 % |
| | | | Yale University | 2 | 4 % |
| | | | Other | 2 | 4 % |
| | | Japan | Keio University | 2 | 4 % |
| | Asia | Turkey | Bogazici University | 1 | 2 % |
| | | Uzbekistan | Samarkand State University | 1 | 2 % |
| | | | *Prefer not to say* | 6 | 12 % |

[†] Other = non-university participants.

## A.2 PLAYER RECOGNIZABILITY

Some experts argue that professional players can indeed be recognized from their moves alone, pointing to recent machine learning studies that achieve high accuracy in attributing games even when results and openings are excluded McIlroy-Young et al. (2021), suggesting that mid- and late-game decisions carry individual traces. Others, however, caution that such recognizability diminishes among elite grandmasters, whose choices converge toward objective best play, making distinctions far less clear. The debate therefore hinges on whether the residual patterns left in high-level games are strong enough to constitute a reliable identity marker, or whether recognizability is largely an artifact of broader repertoires and tendencies observable outside the very top tier. To explore how this issue is perceived in practice, we sought to probe the opinion of our sample by submitting the following question:

> There is a longstanding discussion in chess literature as to whether a player's identity can be inferred from the moves alone. Classical commentators and modern machine learning studies suggest that players exhibit "fingerprints" in their decision making. This raises the question whether recognizability through move patterns is accepted among experts.
>
> To what extent do you agree with the statement:
> *"Professional chess players can be recognized by the moves they play, independently of the final result."*

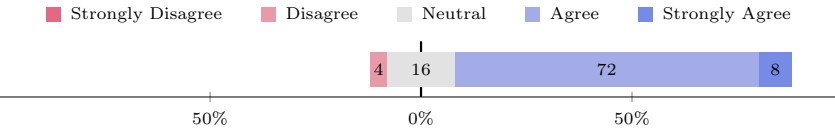

Figure 11: **Distribution of responses to the statement that professional chess players are recognizable by their moves alone.** The horizontal stacked bar represents the proportion of respondents on a five-point Likert scale (from Strongly Disagree to Strongly Agree).

The question was framed to omit any mention of the high accuracy rates achieved by prior AI studies, ensuring that responses would reflect participants' genuine beliefs rather than being primed by this information. The distribution of responses in Figure 11 shows a positive-skewed distribution. A clear majority (80%) agree or strongly agree that professional players can be recognized from their moves. This level of endorsement is considerably higher than expected, given the persistent debate in the community and the presence of skeptical positions regarding the reliability of such recognizability at the elite level. Although a minority of respondents expressed reservations, the overall pattern provides strong evidence-based support for our behavioral stylometry model-based metrics.

The strong consensus on the existence of player recognizability motivates a deeper inquiry into its nature. We therefore posed a follow-up question designed to identify the specific factors that practitioners believe constitute a player's identity. This question was administered to the entire cohort to understand which factors contribute to the definition of a chess persona, independent of whether those factors are ultimately considered strong enough for reliable identification.

> Although many agree that players can be recognized from their games, it is far less clear what exactly makes them recognizable. E.g., what makes Kasparov "Kasparov"? The identity of a chess player appears to be multidimensional, and even experts often disagree on which aspects are most decisive. Understanding which dimensions practitioners themselves consider relevant is crucial for clarifying the concept of "chess persona."
>
> *Which of the following factors, in your opinion, most contribute to making a player recognizable?*

The results in Figure 12 indicate that an inclination toward aggressive or defensive play is perceived as the most defining characteristics, being selected by 88% of participants, respectively. Typical risk management and preferred opening repertoires are also ranked highly, cited by 68% and 60% of the sample, respectively. In contrast, other attributes were considered less significant; characteristic handling of the endgame was endorsed by only 28%, while support for choices between objectively equivalent alternatives (36%) was notably lower than expected (see Appendix A.3). The "Other" category, selected by 12% of participants, captured a range of insightful points. Some respondents leveraged this option to register a premise reject, arguing that recognizability is exceedingly difficult among today's universal top players. Others pointed to more granular factors such as preferences

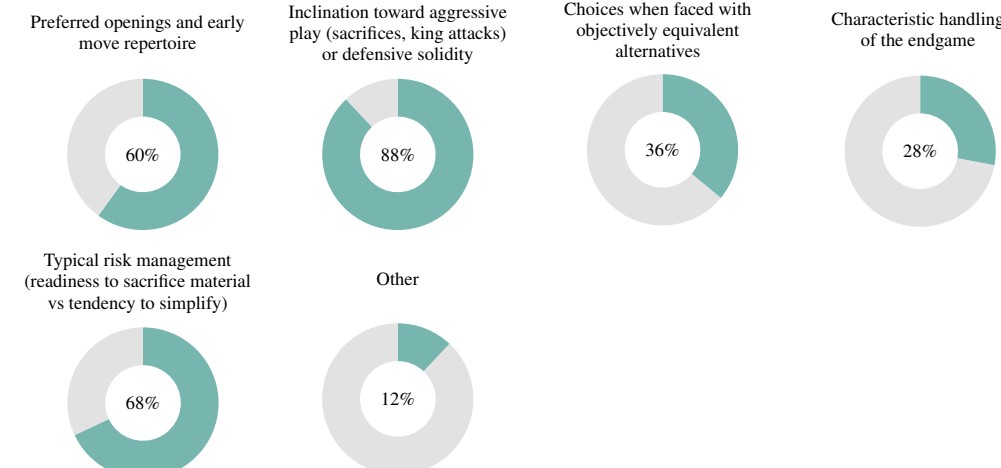

Figure 12: **Perceived contribution of gameplay attributes to player recognizability.** The donut charts display the percentage of respondents who selected each given factor.

for specific pawn structures, weaknesses in opening, middle, and end game. Notably, some participants highlighted time management. This last point is particularly salient; while we concur that decision speed is a powerful discriminative signal, the absence of move-timing information in the PGN datasets used in this work precluded its inclusion in our stylometry model.

## A.3 EXISTENCE AND DEFINITION OF STYLE

Following the question of recognizability, we delve into the related but more fundamental concept of playing style, the existence of which remains a subject of controversy within the chess community. One mindset posits that as players approach optimality, individual style dissolves into a universal pursuit of the objectively best move. A telling case is Anatoly Karpov, who provocatly declared "*Style? I have no style!*," a statement intended to underscore a commitment to pure objectivity. Conversely, the opposing view argues that style is not a deviation from correct play, but rather a discernible pattern of preferences that emerges in complex positions where multiple viable continuations exist. In forced positions with a single correct move, style has no space to manifest; it is in the majority of positions with multiple viable continuations that a player's individuality comes to the fore. Garry Kasparov, offered a paradoxical rebuttal to Karpov's claim, joking that "*His style is precisely to have no style: his essence is to accept only those positions in which there are neither risks nor doubts.*" Even Karpov, indeed, was nicknamed the "boa constrictor" for his recurring board states, and admitted to systematically favoring clear positional lines over tactical complications. This aligns with the long-held idea that style is an expression of personality, as champion Rudolf Spielmann noted in the 1930s: "*Show me your strategic principles in a game and I will tell you who you are.*" This expression is nevertheless constrained by a player's practical abilities and shaped by subjective factors like personal taste (e.g., a kingside attack vs a central buildup) and psychological attitude toward risk. The tension between style as an illusion negated by objective truth and style as a valid construct revealed through subjective choice is critical to our work.

> There is no consensus on whether "style" truly exists in modern chess. Some grandmasters (e.g., Karpov) have claimed they had no style, only the pursuit of objectively best moves; others are consistently described as emblematic of a style. The debate revolves around whether style is an illusion or a legitimate construct. It is also important to distinguish between playing style—broad categories such as "attacking" or "positional"—and persona, the individual identity of a specific player. Two grandmasters may have very distinct personas while still being classified under the same style.
>
> *Do you acknowledge the existence of a playing style in chess, defined as a recurrent pattern of preferences in move selection, or do you believe only the search for the objectively best move matters?*

The results in Figure 13 indicate a near-unanimous agreement among the sampled experts. An overwhelming 92% of participants affirmed that style exists and is identifiable, recognizing differences in preferences and approaches between various players.

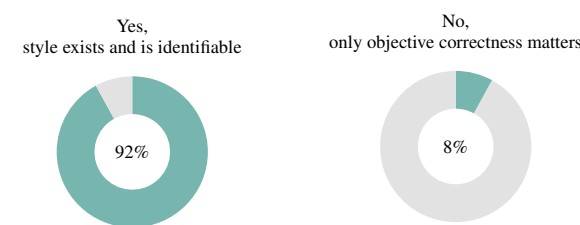

Figure 13: **Expert consensus on the existence of playing style.** Binary question.

We proceed from the premise that style categories are not rigid, mutually exclusive labels but rather useful archetypes for characterizing a player's predominant tendencies. Therefore, we presented our expert sample with a list of the most commonly accepted categories in chess literature. Our goal was to test which of these are broadly considered valid, and to identify whether, in the perception of our respondents, any crucial descriptors were missing from our conventional taxonomy.

> If one accepts that style exists in chess, the next challenge is defining and categorizing it. This is not straightforward: styles may overlap, and manifest differently across contexts.
>
> *Which of the following playing styles do you consider valid and useful categories?*

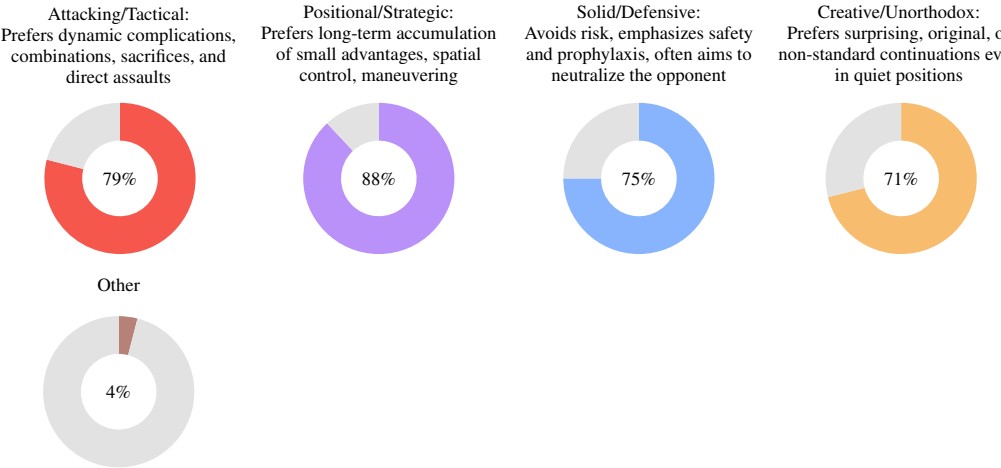

Figure 14: **Validation of conventional playing style categories.** The donut charts show the percentage of respondents who endorsed each of the proposed style categories as valid and useful.

Figure 14 visually summarizes the results. A strong consensus emerged on the validity of conventional style categories, with all four proposed archetypes—Attacking/Tactical (79%), Positional/Strategic (88%), Solid/Defensive (75%), and Creative/Unorthodox (71%)—being widely acknowledged as useful descriptors. Although this confirms the utility of the conventional taxonomy, qualitative feedback from the "Other" category (4%) offered a more nuanced argument. This feedback suggested that more weight should be on the decision-making process rather than the outcomes, arguing that while any strong player can adopt any of the aforementioned styles given the necessity of the position, the true variation arises in how decisions are made. This viewpoint suggests a shift from outcome-based categories to process-oriented ones, framing a player's identity in terms of their characteristic cognitive weighting. If the game process is seen as a product of intuition and calculation, the source of difference between players is the respective weight given to each of these two components. As illustrative examples, Gukesh and Ding Liren were cited as players who rely intensely on calculation, while Magnus Carlsen and Ian Nepomniachtchi were seen as relying more heavily on intuition. This emphasis on the cognitive process directly echoes our earlier point regarding time management as a key dimension for player identification. As noted previously, the time a player allocates to a move is a strong external indicator of their internal decision-making process—crucial information that, while unfortunately unavailable in common PGN datasets, should be a central consideration for future work in this area.

Expert human players demonstrate a remarkable ability to assess complex positions by recognizing abstract visual cues and harmonious piece structures, a skill closely linked to what is often termed "chess beauty." This form of pattern recognition operates on a different level than tactical calculation, relying on an intuitive grasp of a position's strategic potential which is hard to derive from symbolic move notations only. Accordingly, beyond move sequences, a player's identity is often thought to manifest in the visual patterns they characteristically create on the board. A positional player, for instance, might consistently produce games with harmonious piece structures and solid pawn chains, while a tactical player's games may be visually defined by dynamic imbalances and asymmetric configurations. To assess how salient this visual dimension is for our expert sample, we posed the following direct question.

> *Do you believe that visual patterns are important to recognize style?*

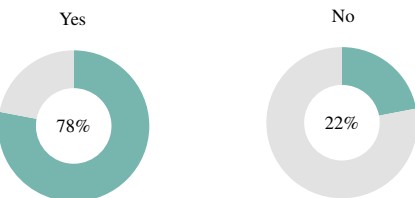

Figure 15: **Perceived importance of visual patterns in style recognition.** Binary question.

As shown in Figure 15, the majority (78%) of respondents affirmed that visual patterns are salient for style recognition. This finding suggests that for the expert community, a player's identity is not solely encoded in symbolic move sequences, but is also tangibly reflected in the characteristic board states and piece configurations they produce. The reported agreement also offers a solid empirical justification for our decision to pioneer a vision-based behavioral stylometry model for chess.

### A.4  STYLE IN GRANDMASTERS

In chess culture, it is common to attribute a dominant style to the great champions of the past: consider the classic contrast between Mikhail Tal, the archetype of the tactical genius who created sacrificial attacks, and Tigran Petrosian, the emblem of prophylactic and defensive play. However, such characterizations, while illustrative, are a simplification. Elite players of any era possess a very broad repertoire, and as experts argue, speaking of "style" at the master level often amounts to highlighting a player's preferences or strengths, but by no means implies they are incapable of excelling in other aspects of the game. This complexity is further deepened by the fact that style is not necessarily a fixed trait. Like any human characteristic, style can evolve with experience: some change their style during their career, others maintain their trademark. This raises a particularly critical question about today's grandmasters, who are often described as "universal." We therefore sought to determine if our expert sample believes that even within this modern paradigm of all-around excellence, it is still possible to attribute a predominant style to a modern elite player.

> At the elite level, players are often described as "universal," capable of playing any type of position well. Yet many analysts argue that even such players retain a dominant style, recognizable across their careers, though it may evolve.
>
> To what extent do you agree with the statement:
> *"Even a modern elite grandmaster, while being nearly universal, still exhibits a dominant playing style."*

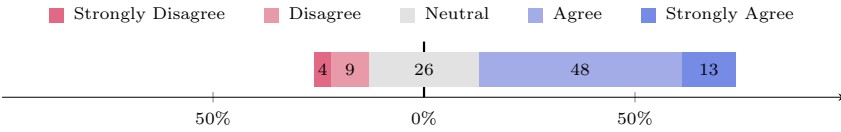

Figure 16: **Distribution of responses to the statement that modern elite grandmasters exhibit a dominant playing style.** The horizontal stacked bar represents the proportion of respondents on a five-point Likert scale (from Strongly Disagree to Strongly Agree).

The expert sample's response to this question, detailed in Figure 16, indicates a prevailing, albeit not unanimous, belief in the persistence of a dominant style. A 61% majority of respondents affirmed this view, supporting the idea that a player's core tendencies remain identifiable even within a universal skill set. The significant 26% of neutral responses, with only a 13% minority in outright disagreement, suggests that the primary source of contention is not whether a dominant style exists, but how to reconcile this concept with the acknowledged versatility of modern players.

To empirically test the practical implications of these beliefs, we transitioned from abstract opinion to a concrete labeling task. We sought to determine whether the majority view—that dominant styles persist in modern grandmasters—is matched by a consistent ability among experts to apply such labels in practice. Participants were therefore asked to assign the previously discussed style categories to each of the GMs who are the subjects of our computational analysis in the main paper. This exercise serves to ground the theoretical discussion, allowing us to measure the degree of consensus that emerges when experts perform this practical classification.

> To empirically ground the discussion, we ask respondents to attempt labeling specific contemporary grandmasters using the style categories introduced above. This helps test whether such labels are perceived as meaningful or not.
>
> *Please assign a dominant style to each of the following grandmasters. If you do not know the player well, or cannot attribute a dominant style, select the appropriate option.*

The results of this practical labeling task, presented in Figure 17, underscore the inherent difficulty of assigning singular style categories to GMs. This challenge is immediately apparent from the "Don't know (cannot assign)" option; it was selected for every grandmaster, representing 19% of responses on average and peaking for Aronian (30%). When a choice was made within the four main style categories, high inter-annotator agreement was observed for only 4 out of the 10 GMs: Anand (attacking), Caruana (positional), Nakamura (attacking), and Vachier-Lagrave (attacking). The remaining GMs received more fragmented and contrasting votes. The "Other" option was leveraged by respondents to provide more specific characterizations. For instance, Carlsen was described as "universal," a label that transcends the given styles. Similarly, Nepomniachtchi was defined through this option as a blend of "creative and aggressive." The use of this free-form option for such prominent players suggests that the conventional taxonomy, while broadly accepted, is sometimes perceived as insufficient to capture the identity of certain top GMs.

## A.5    IMPACT OF CHESS ENGINES AND AI

Our survey concludes by addressing a critical external determinant shaping the concepts discussed thus far: the impact of AI on human play. The perceived rise of the "universal" player and the issues in applying stable style categories are often attributed to the ubiquitous use of chess engines in modern preparation. A central concern within the community is that this technological reliance may be fostering a homogenization of play, eroding the expressive diversity that once defined different eras. We therefore sought to determine whether our expert sample believes such a homogenization is occurring.

> In contemporary practice, the systematic use of chess engines has become virtually indispensable for training, preparation, and post-game refinement. This has raised concern among players and scholars that such reliance may lead to a homogenization of playing behavior: players increasingly converge on the same engine-approved continuations, reducing the expressive diversity once observed across grandmasters of different schools or eras. This risk is considered even more pronounced in chess language models (CLMs). Unlike traditional search-based engines, which aim to compute the objectively best move, CLMs are trained to predict the most statistically likely next move from large corpora of historical games. In other words, they optimize for probability of occurrence rather than chess-theoretical correctness. By reflecting the aggregated tendencies of thousands of players of varied strength, such models might exacerbate stylistic flattening, reproducing the "average" move rather than preserving distinctive personas.
>
> *To what extent do you agree with the following statements:*

The results in Figure 18 confirm the widely held concern that motivates our paper: a strong majority (60%) of respondents agree that intensive reliance on AI has caused a flattening of stylistic diversity among players. This perceived homogenization is particularly noteworthy when contextualized by the second finding, where 70% of respondents affirmed that style would lose its meaning in a hypothetically "solved" version of chess. This result conceptually tethers style to the existence of meaningful human choice and imperfection. In a powerful counterpoint, however, there was unani-

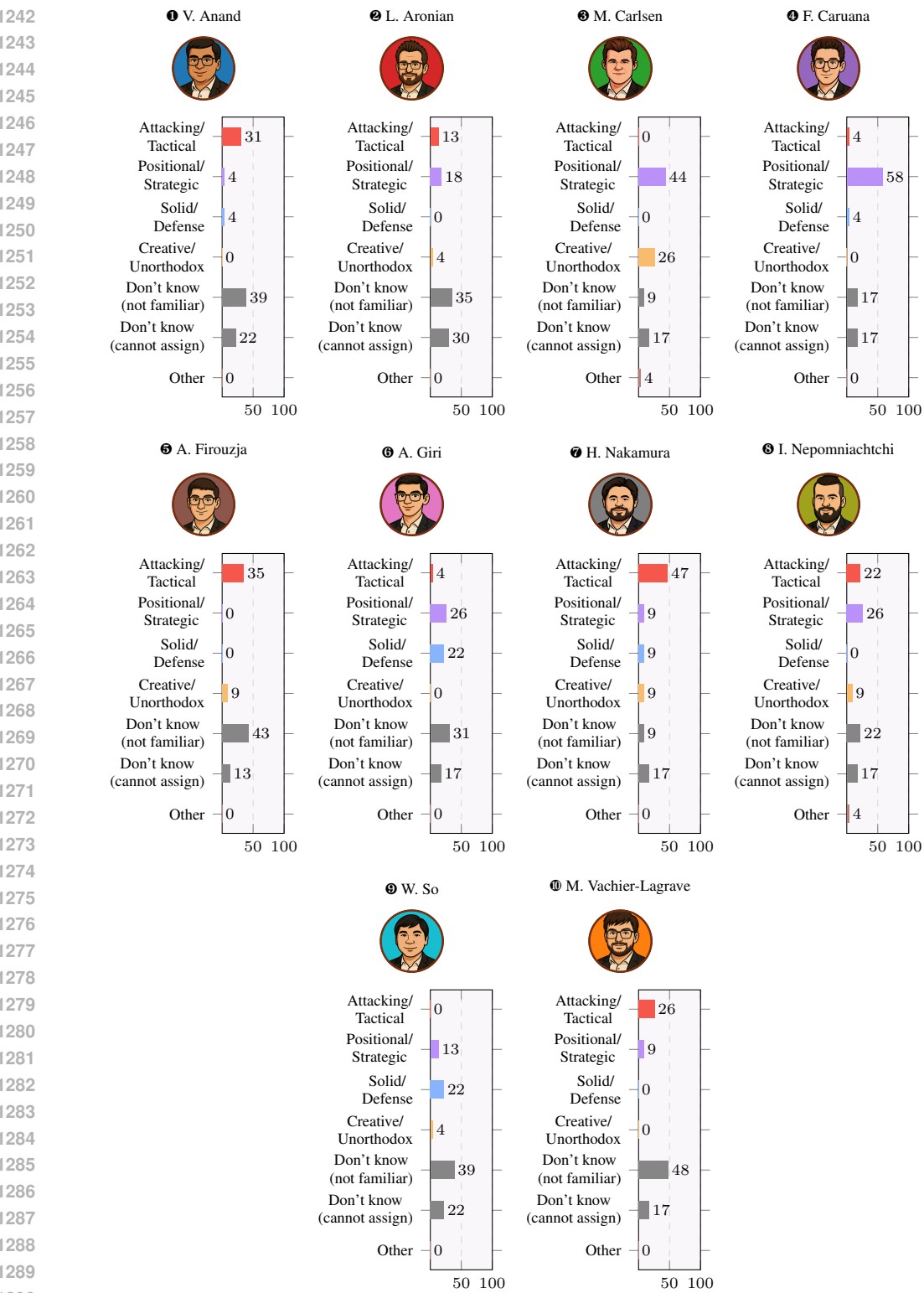

Figure 17: **Distribution of style category assignments for the ten grandmasters featured in this study.** Once choice per grandmaster. Each subplot displays the percentage of respondents assigning a dominant style category to a specific grandmaster.

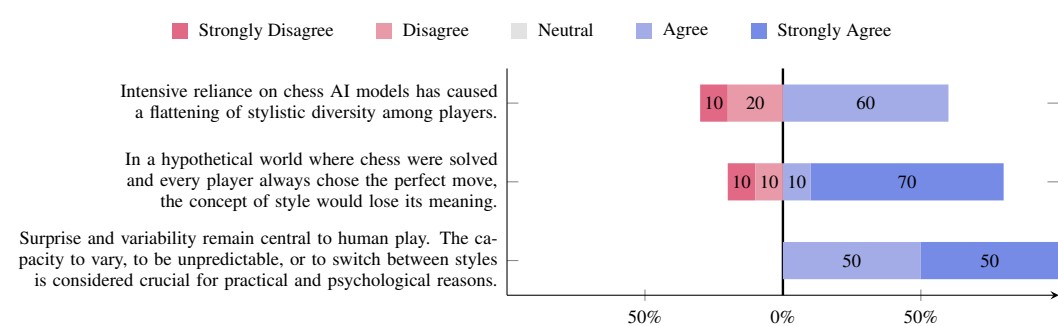

Figure 18: **Perceptions of AI's impact on stylistic diversity and the enduring importance of human variability.** Each horizontal stacked bar represents the proportion of respondents on a five-point Likert scale (from Strongly Disagree to Strongly Agree).

mous agreement (100%) that surprise and variability remain crucial for practical and psychological reasons in human-to-human play.

## B  RELATED WORK

**Chess and AI.**   Abstract games serve as a valuable proxy for real-world skills, providing a rigorous means of evaluating a model's capacities in strategic planning and reasoning, memory and adaptive learning, as well as theory of mind through the inference of an opponent's intent. Chess is a landmark planning problem in AI research, distinguished by a rich history, extensive data corpus, and active community engagement.

*Traditional engines.* Early computer chess relied on heuristics-based techniques, as testified by Turing's initial explorations (Burt, 1955) and implementations like NeuroChess (Thrun, 1994). It culminated in Deep Blue (Campbell et al., 2002) and early versions of Stockfish (Romstad et al., 2008), which used tree-search algorithms with handcrafted evaluation functions. Although most modern engines retain this search-evaluation structure, they have replaced static evaluation with neural networks. In this sense, AlphaZero (Silver et al., 2017) represented a major milestone. It learned to play chess solely through RL from repeated self-play, using a convolution-based residual network to evaluate board positions and to guide Monte Carlo Tree Search (MCTS) in selecting moves. The open-source Leela Chess Zero (Authors, 2018) recreated this approach with key enhancements that accumulated over time, including support for multiple hardware backends, opening rule variants, and the transition to transformer-based architectures.

*Language models.* More recently, chess has been reformulated as a sequence modeling problem because of its text-archived nature. Unlike natural language, chess notations describe a simple, constrained, and deterministic domain. It is important to note a fundamental distinction. Traditional chess engines, whether based on search algorithms or RL, are explicitly optimized to win the game: their architecture supports deep lookahead, evaluation functions, pruning, transposition tables, etc., all tuned to achieve maximal performance in terms of game outcomes. In contrast, traditional language models trained autoregressively to predict the next move lack an explicit representation of the terminal goal (i.e., victory). Their objective is not to win per se, but to maximize the probability of the sequence of moves under construction. Furthermore, their architecture is comparatively simple—not requiring search trees and handcrafted evaluation functions in favor of statistical pattern matching over large game corpora—making them easier to train and adapt. Noever et al. (2020) were among the first to observe that fine-tuned GPT-2 models, under a SSL regime, can generate meaningful moves and plausible strategies. Interestingly, later work demonstrated that even a vanilla GPT model with just 50M parameters, trained from scratch on a few million game transcripts, can achieve a legal move rate of 99.8% and ∼1,300 Elo—without signs of memorization (Karvonen, 2024). Foundation models have expanded the scope of chess AI, powering rule induction (DeLeo & Guven, 2022; Stöckl, 2021), move quality assessment (Kamlish et al., 2019), state tracking (Merrill et al., 2024; Toshniwal et al., 2022), vision-based playing (Czech et al., 2023), commentary generation (Lee et al., 2022), and auxiliary generative tasks (Feng et al., 2023). Fine-tuning on chess textbooks, commentary, and tactical calculations has also proven effective (Alrdahi & Batista-Navarro,

2023; Feng et al., 2023; Wang et al., 2025), giving the model both move sequences and explanatory texts. To maximize performance, Ruoss et al. (2024) abandoned SSL and distilled Stockfish into large-scale decoder-only transformers via supervised learning on engine-crafted annotations, reaching an impressive 2,895 Lichess blitz Elo against humans. They trained transformers with up to 270M parameters to predict action-values[5] given a board state, using 10M chess games annotated by Stockfish 16. In parallel, Monroe & Team (2024) (Leela Chess Zero team) introduced Chessformer, an encoder-only transformer architecture with chess-specific optimizations for action-value estimation. After supervised training on AlphaZero self-play data, models (up to 240M) further surpassed Ruoss et al. baselines while using fewer FLOPs. Notably, Zhang et al. (2024) provided evidence that self-supervised generative models can attain Elo ratings beyond those of any player in their training corpus—a phenomenon known as transcendence. LLMs have also shown surprising zero-shot chess ability, which motivated inclusion in evaluation suites like BIGBench (Srivastava et al., 2023). Carlini (2023) proved that the accuracy of LLMs in solving chess puzzles decreases by more than half when the PGN move history provided as context differs from the actual game from which the puzzle position was extracted. This result highlights the adaptability of chess language models to the inferred playing strength: sequences of weaker moves bias the model toward imperfect continuations, whereas sequences of stronger moves guide it toward more accurate play. Zhang et al. (2025b) fine-tuned Open-LLaMA-3B to generate the best move in Standard Algebraic Notation (SAN) from a given board state in Forsyth-Edwards Notation (FEN). By annotating training data with Stockfish and leveraging high-depth searches in its alpha–beta tree, they achieved an Elo rating of 1,788. More recently, Kaggle's Game Arena, in partnership with Google DeepMind, hosted a text-only chess tournament where eight commercial LLMs competed in a bracket format (best-of-four over three rounds), with live commentary from Hikaru Nakamura, Magnus Carlsen, and GothamChess.[6] These successes have sparked a research agenda centered on interpreting superhuman models by probing their internal representations, revealing chess concepts and look-ahead in AlphaZero (Jenner et al., 2024; McGrath et al., 2021), and board state encoding in self-supervised language models (Karvonen, 2024). From an input perspective, exclusive use of FEN is typical only for engine distillation procedures (Ruoss et al., 2024; Monroe & Team, 2024; Zhang et al., 2025b), where the focus is on evaluating static states or targeting Chess960 puzzles that randomize the back-rank starting position. When the goal is to model full games move by move, the progressive history in PGN format becomes essential.

Our work builds on small chess language models trained with *both SSL and RL (GRPO)*–PGN input, focusing not on Elo gains but on exploring, for the first time, chess MoEs with player-personalized experts. The intersection of chess and GRPO remains under-explored, with most efforts centering on reasoning LLMs that output situation analyses other than suggested moves. Chen et al. (2025) fine-tuned Qwen-2.5-7B-Instruct with GRPO for Xiangqi (Chinese chess), using combined PGN and FEN inputs alongside multi-dimensional rewards designed to improve both output format and engine-evaluated quality. Hwang et al. (2025) applied GRPO to fine-tune Qwen-2.5 and LLaMA-3.1 models for chess puzzle solving, where the input representation was restricted to FEN and the reward signal came solely from engine-derived post-move win probabilities. By contrast, our work investigates GRPO in the context of traditional chess with small, non-reasoning language models, employing legality-based reward strategies and analyzing their effects relative to the SSL-only stage.

**Human-AI alignment in chess.** Humans engage with chess AI both as competitors and training partners. This has motivated research aimed at predicting the moves some humans are likely to make, rather than those that are strictly optimal. Dealing with bots with contrasting styles trains the user to recognize and respond to different strategies, so they allow to exercise cognitive flexibility. Commercial products like Play Magnus and Chess.com's bots are player-personalized, though their methods remain undisclosed. In open research, McIlroy-Young et al. (2020) developed Maia, a supervised adaptation of Leela Chess Zero that predicts moves of average human players at specific rating levels, with separate models trained per Elo band. Maia-2 moved to an efficient and unified model using skill-aware attention (Tang et al., 2024). Closer to our work, the same authors proposed models capable of identifying hundreds of individual players from their move patterns: first by fine-tuning Maia-1900 (McIlroy-Young et al., 2022), then by training a vision transformer from

---

[5]In this context, each legal move is an action, the value is the quality or expected return of an action; these predictions can be used to build a chess policy, which is a probability distribution over all legal moves that reflects the model's belief about how likely each move is to be the best choice in a given position.

[6]https://www.kaggle.com/benchmarks/kaggle/chess-text/tournament

scratch on sequences of moves represented as 3D tensors with human-engineered features (McIlroy-Young et al., 2021). Influenced by this research line, we create individual experts and implement a model-based behavioral stylometry metric, eliminating the need for feature engineering by operating directly on *raw game video recordings*. For context, McIlroy-Young et al. (2021) trained on 10K to 40K games from 16K players of Elo 1,000-2,000 and only 1.8K high-ranked players selected from lichess and chess.com leaderboards. Instead, we target 10 GMs with an average 2,816 Elo, and train vision foundation models to obtain appreciable performance with few training data, 1,000 games.

**Expert merging.** Growing evidence suggests that diversity beats strength. Dobre & Lascarides (2017) proved the utility of MoE in complex games such as Settlers of Catan, where experts were trained on diverse datasets. Heterogeneous teams of Go agents have been shown to outperform solitary agents (Kakade & Langford, 2002) and homogeneous teams (Marcolino et al., 2014). Helfenstein et al. (2024) explored chess MoEs, but they used external MCTS and specialized by game phase rather than *player behavior*.

## C  REPRODUCIBILITY

### C.1  DATASET DETAILS

The dataset was constructed by merging PGN game files from three sources of grandmaster-level games: PGNMentor,[7] Chess.com,[8] Lichess.[9] Following the merge, we applied filtering to remove duplicate games, entries with malformed PGN formatting, and quality glyphs such as "?!".

**Color balancing**  Since our models are trained exclusively on moves from individual masters, we performed color balancing within each master's game collection. For each player, we downsampled games of the overrepresented color (White or Black) to match the count of the underrepresented color, ensuring equal representation of both colors in the training data.

**Data augmentation via mate completion**  To increase the representation of complete mating sequences in the dataset, we performed an additional augmentation step. Grandmaster games typically end in resignation before the actual mate is delivered, resulting in a dataset deficient in explicit checkmate patterns. This augmentation aims to help trained models learn to execute legal mating sequences while preserving grandmaster playing style, which typically favors the shortest possible mate. For each game ending without checkmate, we used Stockfish to analyze the final position and determine whether a forced mate existed within 10 moves. When such a mate was detected, we extended the PGN by appending the shortest available mating sequence.

**Video generation**  Frames for video generation, which represent the evolving board state and highlight the last GM's move, were automatically generated from PGN strings using the `python-chess` (v1.11.2) library.[10]

**Grandmaster statistics**  Research on human chess players indicates that while novices explore a wide breadth of openings, experts tend to specialize by employing a preferred repertoire. Concurrently, GMs compensate for a narrower diversity of initial moves with a greater depth of variation within their chosen opening systems (Barthelemy, 2025). Our analysis confirms these patterns in the selected GMs (Table 4). Their opening frequencies reveal a distinct hierarchy: each player's repertoire is generally dominated by two or three primary systems, while secondary options are employed at logarithmically lower frequencies. Openings were identified using the Lichess Encyclopedia of Chess Openings, which defines a vocabulary of 496 classes.[11] Despite this large vocabulary, collective specialization at the highest level is remarkably concentrated: the top-5 openings across all GMs are drawn from only 7 unique variants.

---

[7]https://www.pgnmentor.com/

[8]https://www.chess.com/games, covering games from January 2015 to June 2025.

[9]https://huggingface.co/datasets/Lichess/tournament-chess-games

[10]https://python-chess.readthedocs.io/en/v1.11.2/

[11]https://github.com/lichess-org/chess-openings

**Licenses** Lichess releases its data under the Creative Commons CC0 license. PGNMentor and Chess.com publicly release large collections of games for free download, and their data are widely used in academic research (Burduli & Wu, 2023; Adnan et al., 2024; Bonato & Walaa, 2025).

Table 4: **Statistical overview of the grandmaster persona datasets (part 1).** The columns detail, from left to right: (i) the distribution of game results stratified by color; (ii) the frequency of the top-5 openings stratified by color; (iii) the provenance of PGN games.

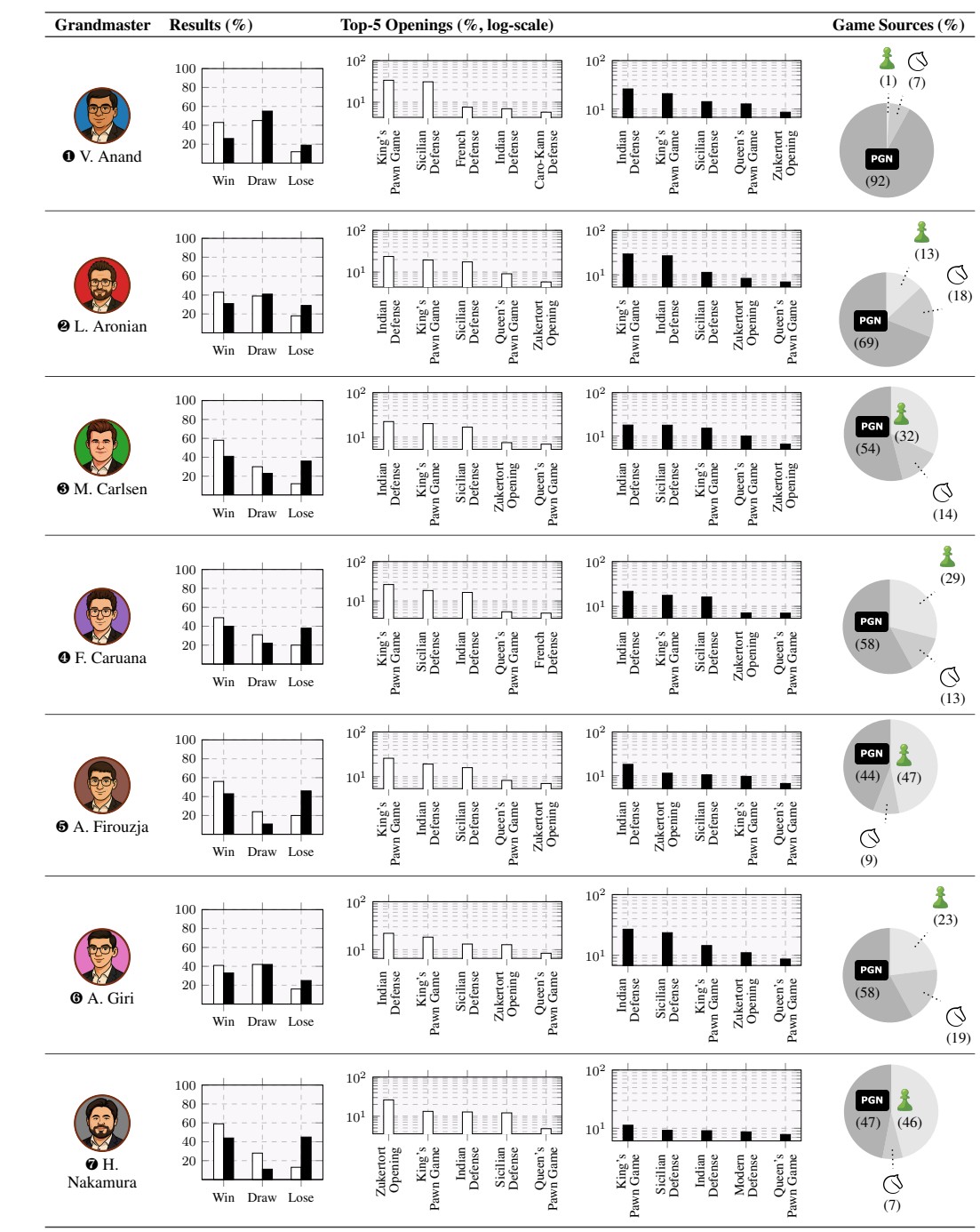

Table 5: **Statistical overview of the grandmaster persona datasets (part 2).**

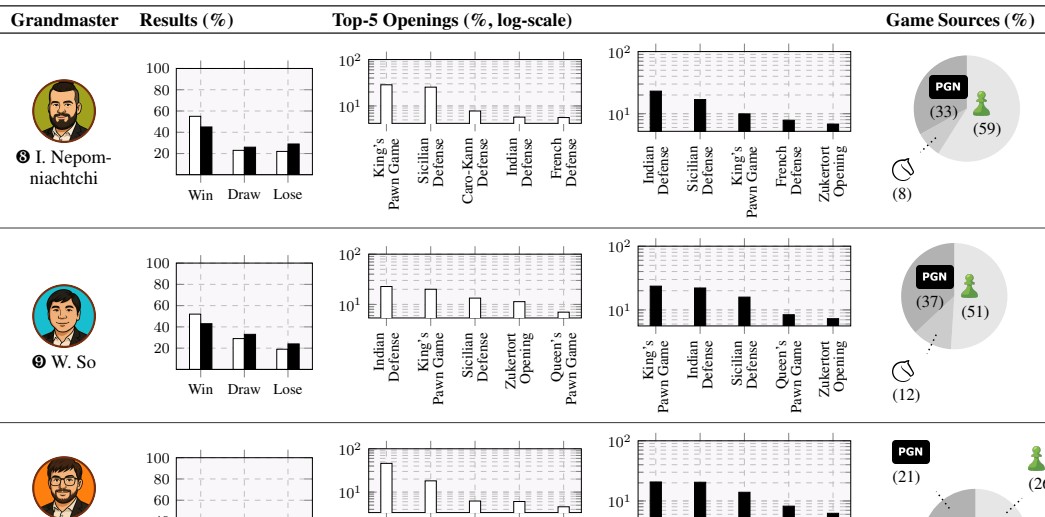

Table 6: **MOM vocabulary.** A 32-token character set containing all necessary letters, digits, and symbols to describe move sequences within PGN games.

| Category | # Tokens | Tokens | Comment |
|---|---|---|---|
| Files | 8 | a b c d e f g h | Board columns |
| Digits | 10 | 0 1 2 3 4 5 6 7 8 9 | Board rows (ranks) and move numbers |
| Pieces | 5 | K Q R B N | King, queen, rook, bishop, knight – pawns have no letter in SAN |
| Castling | 1 | O | King-side (O-O) and queen-side (O-O-O) castling |
| Move | 5 | x + # = - | Capture, check, checkmate, promotion, and castling dash |
| Separators | 3 | . ⎵ ; | Period after move numbers, space separator between moves, special delimiter denoting game start/end |

## C.2 IMPLEMENTATION AND EVALUATION DETAILS

**Tokenizer details** To ensure a lightweight and parameter-efficient model architecture, we selected a minimal 32-character vocabulary (Table 6), the most compact set necessary for representing PGN sequences. This decision is directly informed by Karvonen (2024), who demonstrated that employing the GPT-3.5's default BPE tokenizer with 50,257 entries would inflate the model's parameter count by 25M. Furthermore, Karvonen's analysis revealed that the larger tokenizer provides no commensurate improvement in encoding efficiency for this domain, as it already encodes PGN strings with slightly over 1 character per token (excluding spaces). Accordingly, all our experiments were conducted using seed models that rely exclusively on this tokenization scheme. In line with Karvonen (2024) and Zhang et al. (2024), we ensured that–during training–every batch began with the sequence ";1." to serve as a delimiter for a new game.

**Evaluation details** Automatic legality checks on PGN strings (i.e., game validation) were performed using the `python-chess` (v1.11.2) library.

**Hyperparameters** All models were trained using hyperparameters optimized through Gaussian process-based Bayesian optimization for the most critical parameters, with remaining settings determined via standard search methods. The optimization ranges and final selected configurations are presented in Tables 7 and Table 8.

---

**SSL/RL Hyperparameter Setting**

---

**SSL Training configuration:**
lr $= [1e-7\ldots2e-6^*\ldots1e-4]$, weight decay $= [1e-5\ldots1e{-}4^*\ldots1e-1]$
dropout $= 0.0$, batch size $= 8$
warmup steps $= 600$, training steps $= 6000$
seed $= 960$
**RL Training configuration:**
lr $= [1e-7\ldots6e-7^*\ldots1e-4]$, $\beta = [0.01\ldots0.06^*\ldots0.1]$
group size $= 8$, batch size $= 64$
warmup steps $= 1000$, training steps$=10000$
seed $= 960$

---

Table 7: **SSL and RL hyperparameter sweep.** Ranges come from the optimization config; starred values correspond to the selected training setting.

---

**Stylometry Hyperparameter Setting**

---

§ **Pretraining hyper-parameters**
**Training configuration:**
lr $E_\psi = [1e-7\ldots1e{-}6^*\ldots1e-3]$, lr classifier $= [1e-5,\ldots1e{-}4^*\ldots1e-2]$
weight decay $= [1e-5\ldots1e{-}4^*\ldots1e-1]$, dropout $= [0.15\ldots0.3^*\ldots0.4]$
batch size $= 92$, pos weight $= 30$
epochs $= 40$, warmup steps $= 1500$, training steps$=15000$
classifier hidden dim $= 256$
seed $= 960$

§ **Finetuning hyper-parameters**
**GE2E Loss parameters:**
$\lambda_m = [0.05\ldots0.8^*\ldots1.0]$
$\lambda_c = [0.001\ldots0.7^*\ldots1.0]$
$\mu = [0.1\ldots0.5^*\ldots1]$
**GE2E parameters:**
$W = [1.0\ldots8.5^*\ldots15.0]$
$b = [-12.0\ldots-10^*\ldots2.0]$
**Training configuration:**
lr $= [1e-7\ldots5e{-}6^*\ldots1e-3]$, weight decay $= [1e-6\ldots2e{-}3^*\ldots1e-1]$
dropout $= [0.05\ldots0.15^*\ldots0.4]$
batch size $= 4$, $N = 5$, $M = 10$, $F = 5$
epochs $= 20$, warmup steps $= 2500$, training steps$=25000$
LSTM hidden dim $= 512$
seed $= 960$

---

Table 8: **Stylometry hyperparameter sweep.** Ranges come from the optimization config; starred values correspond to the selected training setting.

**Compute resources**  All experiments were performed on a workstation running Ubuntu 20.04.3 LTS, equipped with an Intel® Core™ i9-10900X CPU @ 3.70GHz and 128GB of RAM. The optimization of behavioral stylometry models was conducted on two NVIDIA GeForce RTX3090 GPUs (24GB VRAM), while all remaining computations were executed on a NVIDIA GeForce RTX5090 (32GB VRAM).

## D   MERGING TECHNIQUES

A potential consequence of weight interpolation in MoM stitching is the catastrophic forgetting of acquired chess capabilities. To determine the optimal merging configuration for downstream performance, we systematically evaluate diverse parameter consolidation techniques spanning weight-based, gradient-informed, and subspace-oriented approaches over the fully merged model.

Weight-based methods form the foundation of our analysis, with naive averaging (Wortsman et al., 2022) serving as the baseline approach, which assigns uniform importance to all parameters. Task arithmetic (Ilharco et al., 2023) provides a more principled alternative by leveraging task-specific weight differences relative to the base model, thereby preserving specialized capabilities during integration. To capture higher-order parameter relationships, we evaluate KnOTS (Stoica et al., 2025), which employs Singular Value Decomposition to identify and merge critical parameter subspaces that simpler averaging methods might compromise.

Beyond weight-centric methodologies, we investigate gradient-based approaches utilizing Fisher information matrices (Matena & Raffel, 2022; Lee et al., 2025). These methods approximate the Hessian of the loss function to weight parameters according to their empirical importance in the optimization landscape, with parameters exhibiting higher Fisher information values receiving proportionally greater influence during consolidation. This information-theoretic paradigm fundamentally differs from uniform weighting by prioritizing parameters that contribute most significantly to model performance.

Through a systematic comparison of these techniques across 10 experimental runs involving 300 games against Stockfish level 0, we assess both the performance preservation and the statistical significance of the observed differences. As shown in Figure 19, Fisher-based merging achieves the highest win rate (57.7%) with minimal performance degradation (-3.3% relative to the best individual expert), while naive averaging maintains substantial performance (51.7%) with moderate degradation (-9.3%). Wilcoxon signed-rank tests confirm that these performance differences are statistically supported across our experimental runs.

Notably, the Fisher method—aptly named for our chess domain—demonstrates superior resilience to parameter consolidation. However, for our primary analysis, we employ naive averaging despite its slightly lower performance. This choice preserves the fundamental principle of equal expert contribution, as our analysis revealed that Fisher's optimal performance was achieved through biased master contributions, thereby contradicting the democratic nature of mixture-of-experts architectures that constitute the core methodological contribution of our work.

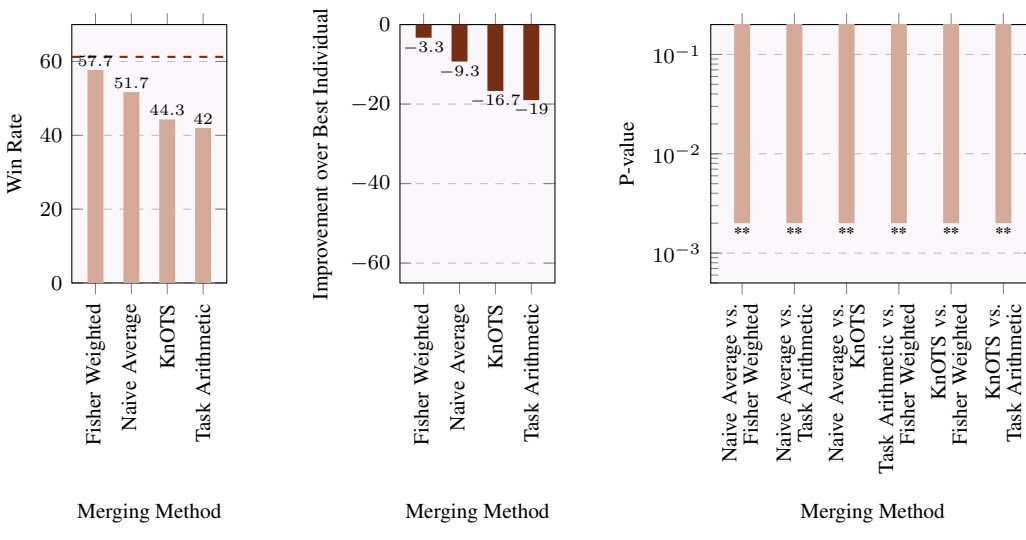

(a) Win rate comparison.

(b) Win rate improvement of the merged models over the best individual model.

(c) Statistical significance of pairwise comparisons (Wilcoxon signed-rank test).

Figure 19: **Win Rate comparison between merging algorithms.**