# OpenReview forum: "Mixture of Masters: Sparse Chess Language Models with Player Routing"
_ICLR.cc/2026/Conference — Submitted to ICLR 2026_

### Official Review · Reviewer_Rgkk · 2025-10-31

**Soundness:** 3
**Presentation:** 4
**Contribution:** 2
**Rating:** 6
**Confidence:** 4

**Summary:**

The paper shows that it is possible to combine multiple LLM style models via MOE to get a more powerful* chess engine than the equivalent non-MOE. They show that their experts when trained on different datasets behave differently via an interesting vision based classifier. Along with some analysis of using RL to train the underlying LLMs and interpreting the different experts.

*The definition of power is not the standard method (Elo/Glicko) used in chess.

**Strengths:**

It's an interesting result for the chess community and while LLM for chess is not a novel idea their RL methods and MOE are. The attempts at interpreting the models are also strong. The choice of a smaller LLM instead of a full GPT-2 model is also refreshing.

**Weaknesses:**

I find parts of this paper hard to follow as the authors use terms from LLM training to describe process that RL/chess AI researchers have been doing for decades. Why are you using GRPO? How does it compare to methods used to train much stronger models like mini-max (stockfish) or UCT(Alpha Zero Chess)? They also don't provide Elo or clear direct comparisons between the models, using FIDEscore instead.

The use of text to represent games was never discussed, why is a chess model getting less than 100% accuracy on picking a legal move OK?

My main issue with this paper is that it is unfocused. I don't see a strong direction to go with it. It's not very useful to LLM people as the methods are highly specialized to chess, and they don't clearly explain what chess or the traditional RL community can gain from it.

I'm giving this a weak accept as I think it's an interesting paper, but I am not confident on it's long term impact.

**Questions:**

What is the Elo of your models? Can you run a tourmanment with the different instantiations and Stockfish, then report the Elo of each model. I'd recommend starting all models at 1500, and using an deviation of 350, then using Glicko 2 for the Elo updates.

Can you explain how this is self-supervised learning? To me it looks like a standard supervised learning approach.

What does "decentralized" and "compositional" mean in this paper?

Why was FIDEScore used instead of raw wins? What is the draw rate in your results?

What is the takeaway for a chess engine builder? Is this a way to outperform stockfish for the same compute? More interpretable? More human?

---

> ### Author Response · Authors · 2025-11-27
> **Answer to Reviewer Rgkk [1/5]**
>
> We sincerely thank the reviewer for the thoughtful feedback and for recognizing the novelty of our RL methods, MoE approach, and interpretability contributions. We address each concern below with clarifications and additional context from our revised manuscript.
>
> ## Weakness 1: Terminology, GRPO Justification, and Evaluation Metrics
>
> > [Reviewer] "Why are you using GRPO? How does it compare to methods used to train much stronger models like mini-max (stockfish) or UCT (Alpha Zero Chess)?"
>
> > [Reviewer] "They also don't provide Elo or clear direct comparisons between the models, using FIDEscore instead."
>
> **Distinction from traditional chess engines:**
> We acknowledge the potential confusion and clarify that our approach represents a fundamentally different paradigm from traditional chess engines. State-of-the-art engines like Stockfish (minimax with alpha-beta pruning) and AlphaZero (UCT-based MCTS with neural network evaluation) are explicitly optimized to **win games** through deep lookahead, evaluation functions, pruning strategies, and transposition tables. In contrast, chess language models—including ours—are trained to **predict the next move** by maximizing the probability of move sequences under autoregressive generation. Our architecture lacks explicit search mechanisms and does not represent terminal goals (victory) in its objective function. Instead, it performs statistical pattern matching over game corpora in PGN format.
>
> Our work is framed within the emerging research area that explores chess as a structured domain for assessing the generalization, planning, and reasoning abilities of large language models. Instead of competing with traditional engines, our aim is to investigate how language-model–based architectures develop and demonstrate strategic skills when limited to next-move prediction, and to determine the extent to which these models can show planning-like behavior without explicit search mechanisms.
>
> **GRPO justification:**
> Traditional RL algorithms (e.g. those used in AlphaZero) are incompatible with our text-generation domain because they assume access to explicit value functions, search trees, and deterministic state transitions—none of which apply to autoregressive language modeling. GRPO (Group Relative Policy Optimization) is specifically designed for next-token prediction tasks and addresses a critical limitation we observed: **SSL-only training induces memorization rather than generalization**, leading to illegal move generation due to overfitting and distributional shift (Lines 142–145, 157–161).
>
> As detailed in our response to Reviewer `HMfV`, GRPO substantially improves legality by incentivizing move exploration and enabling the model to identify multiple valid continuations from a given position, something SSL alone cannot achieve. Within the chess language modeling domain our work is the **first to apply RL-based text generation specifically for move generation** with legality-focused rewards (Lines 1379–1388).
>
> **Evaluation metrics:**
> We respectfully clarify that FIDEScore, WinRate, and DrawRate are **standard proxies for model performance** in the chess language modeling literature. These directly address the reviewer’s concern about the lack of Elo reporting and the use of FIDEScore. These metrics are reported because:
> 1. They provide interpretable, fine-grained insights into model behavior across multiple dimensions,
> 2. They align with evaluation protocols established in prior work on chess LLMs, and
> 3. While Elo/Glicko-2 ratings are valuable for ranking systems, they introduce complications inappropriate for our experimental design. These metrics require extensive round-robin tournaments to achieve rating convergence and are inherently relative—dependent on arbitrary initialization choices and only meaningful when comparing many models against each other. In contrast, we focus on controlled comparisons against a fixed, reproducible opponent (Stockfish at specified levels), providing direct interpretability. Our evaluation scale (10 runs × 100 games = 1,000 games per configuration) exceeds established protocols: Zhang et al. (2024, Transcendence) used 100 games total, while Karvonen (2024) reported similar sample sizes. FIDEScore aggregates WinRate and DrawRate using tournament-standard weighting (1 point for win, 0.5 for draw, 0 for loss), providing a single interpretable performance measure. We have expanded Table 2 to report WinRate and DrawRate separately, with a detailed discussion of how GRPO affects both metrics (see response to `HMfV`, new RQ3).

---

> ### Author Response · Authors · 2025-11-27
> **Answer to Reviewer Rgkk [2/5]**
>
> ## Weakness 2: Text Representation and Legality
>
> > [Reviewer] "The use of text to represent games was never discussed… why is a chess model getting less than 100% accuracy on picking a legal move OK?"
>
> **Text representation justification:**
> The use of text-based PGN representation for chess is well-established in recent literature. Karvonen (2024) and Zhang et al. (2024) demonstrated that even vanilla GPT models trained from scratch on PGN transcripts can achieve strong performance (~1,300 Elo) without memorization, while models like GPT-3.5 exhibit surprising zero-shot chess ability. PGN provides a natural, human-readable encoding that preserves move sequences and is compatible with standard LLM architectures, making it the de facto standard for chess language modeling when the goal is to model full games move-by-move rather than evaluate static board states (Lines 128–130, Appendix B).
>
> **Legality clarification:**
> The legality percentage represents the **percentage of complete games where the model generated zero illegal moves**, not the rate of legal moves over all generated moves (Lines 327-329). Given \(N\) evaluation games, if \(M\) games end due to illegal move generation, legality \(L = 1 - (M/N)\). **Therefore, 80% legality score means that in 80 out of 100 games, the model completed an entire game without generating any illegal moves.**
>
> **Unconstrained autoregressive language models have no inherent mechanism to enforce legal move generation**, as the output vocabulary spans all possible character sequences, not just valid SAN moves from the current position. Unlike chess engines with explicit board representations and legal move generators, LLMs must learn rule adherence purely from pattern recognition over training data, making illegal move generation an expected failure mode that we explicitly measure and address through GRPO.
>
> This issue is well known not only in related works in the literature but also in production systems. For instance, in the Kaggle Game Arena tournament, where in October 2024 state-of-the-art LLMs like Gemini 2.0 Flash and GPT-4o were tested in live chess matches, organizers employed a **multi-trial mechanism allowing up to three retries** before forfeiting the game to mitigate the hallucination issue. Alternative mitigation strategies include constrained decoding to force legal SAN outputs (e.g., beam search restricted to legal move vocabulary).
>
> We emphasize that our **zero-shot, no-retry evaluation** provides a more rigorous test of the model's intrinsic rule adherence: we employ a **strict no-retry policy** where a single illegal move results in immediate forfeiture, and we use greedy decoding to eliminate sampling variance (Lines 320–322). Under this rigorous protocol, our legality scores **consistently surpass state-of-the-art open-source chess language models** used as seed checkpoints (Lines 411–413). To allow qualitative verification, we have deployed our model as a **playable Lichess bot** (https://lichess.org/@/mixture-of-masters, link in anonymous repository), demonstrating legal play in real time.

---

> ### Author Response · Authors · 2025-11-27
> **Answer to Reviewer Rgkk [3/5]**
>
> ## Weakness 3: Paper Focus
>
> > [Reviewer] "My main issue with this paper is that it is unfocused. I don't see a strong direction to go with it. It's not very useful to LLM people… and they don't clearly explain what chess or the traditional RL community can gain from it."
>
> We respectfully disagree with the assertion that the paper lacks focus. The work is centered on a clear and coherent objective: understanding how **modular, expert-specialized language models** can acquire, express, and preserve strategic and stylistic competencies in a highly structured decision-making domain. This yields **three concrete and interconnected contributions** to both the chess AI community and the broader LLM community:
>
> 1. **First chess MoE with player-personalized experts (Sections 3.1–3.2).**
>    We show that independently trained GM-emulating experts can be composed into a sparse MoE system that outperforms each individual model while preserving stylistic diversity,a property repeatedly highlighted as essential by professional players (Appendix A). This contrasts with standard Mixture of "Experts" approaches in the LLM community, where experts are not designed to exhibit particular specialization. Our setting, where experts are *truly distinct* in both data distribution and behavioral intent, provides one of the clearest demonstrations to date of the benefits of authentic specialization and compositionality.
>
> 2. **A domain-specific focus that is consistent with—and builds upon—prior influential work, while yielding general insights.**
>    Similar to prior research that has used chess as a rigorous testbed (e.g., studies of LLM reasoning, planning, and scaling behavior), our work concentrates on a single domain to enable controlled evaluation and interpretability. Importantly, the methodological insights extend beyond chess. Our results show that modularity, persona-conditioned routing, and compositional inference can improve reasoning and decision prediction in settings where traditional LLMs struggle to preserve diversity or avoid mode collapse. Recent work such as *Transcendence* (NeurIPS 2024) has shown how domain-constrained studies on chess can reveal scalable behavioral patterns in LLMs; our findings provide a complementary perspective grounded in strategic, sequential decision-making.
>
>    Thus, while chess provides a concrete evaluation environment, the underlying lessons—on specialization, routing, knowledge sharing, and the limits of dense monolithic models—generalize to broader LLM applications.
>
> 3. **Novel vision-based behavioral stylometry (Section 3.3).**
>    We introduce a video-based style-identification pipeline that operates directly on raw visual game recordings, removing the need for hand-engineered features. Beyond chess, this establishes a general framework for learning behavioral signatures from multimodal sequential data, potentially applicable to domains such as speaker identification from video, player modeling in esports, or fine-grained multimodal preference modeling.
>
> 4. **First application of RL (GRPO) to move generation in chess LLMs (Section 3.1).**
>    We demonstrate that GRPO mitigates illegal-move memorization while enhancing strategic fidelity -- a concrete application of RLHF-style techniques to a structured prediction domain. This offers insight for the RL community into how preference-based optimization interacts with long-horizon sequential outputs beyond natural-language tasks.
>
> Taken together, these contributions articulate a unified direction: **leveraging modular, expert-grounded architectures to advance strategic reasoning, specialization, and stylistic diversity in language-model–based systems**. The takeaway for the LLM community is that compositional MoE architectures can meaningfully preserve diversity and interpretability in specialized domains. The takeaway for the chess community is the introduction of new tools for building transparent, style-aware engines.
>
> In this sense, the paper is focused: it uses chess as a principled testbed to demonstrate generalizable mechanisms for specialization, reasoning, and modularity in language models.

---

> ### Author Response · Authors · 2025-11-27
> **Answer to Reviewer Rgkk [4/5]**
>
> ## Question 1 & 4: Elo Ratings and FIDEScore
>
> > [Reviewer] "What is the Elo of your models? Can you run a tournament and report the Elo of each model…?"
>
> > [Reviewer] "Why was FIDEScore used instead of raw wins? What is the draw rate in your results?"
>
> Please see our response to Weakness 1 above. WinRate and DrawRate serve as direct, interpretable proxies for performance, consistent with established chess LLM evaluation protocols. FIDEScore aggregates these metrics using tournament-standard weighting. We have expanded Table 2 to report all three metrics separately, with detailed analysis of GRPO's effects on playing style (RQ3: GRPO increases DrawRate across all experts, indicating a shift toward defensive, safety-oriented play).
>
>
> ## Question 2: Self-Supervised Learning
>
> > [Reviewer] "Can you explain how this is self-supervised learning? To me it looks like a standard supervised learning approach."
>
> This terminology follows the established convention in both the general LLM literature and chess language modeling specifically. During the SSL phase (prior to RL), the model undergoes standard autoregressive language model training via next-token prediction on PGN strings without additional labels (Lines 136–141). While this could be characterized as supervised fine-tuning (SFT) in some contexts, the broader LLM community and chess-specific works including Karvonen (2024) and Zhang et al. (2024) consistently use "self-supervised learning" to distinguish this approach from methods that incorporate external annotations (e.g., engine evaluations or move quality labels in chess; human feedback or task-specific labels in general NLP). We adopt this nomenclature for consistency with the domain literature.
>
>
> ## Question 3: Decentralized and Compositional
>
> > [Reviewer] "What does 'decentralized' and 'compositional' mean in this paper?"
>
> "**Decentralized**" refers to our independent, asynchronous training of experts (Lines 96–102), where each GM model is trained in isolation without cross-contamination, enabling parallel development and modular updates. "**Compositional**" describes the MoE stitching process (Section 3.2), where specialized experts are combined into a unified model that dynamically routes between personas based on game state. Together, these terms emphasize our framework's modularity and scalability—new experts can be added without retraining the entire system, and the routing mechanism provides interpretable attribution of decisions to specific playing styles.
>
>
> ## Question 5: Practical Takeaway for Chess Engine Builders
>
> > [Reviewer] "What is the takeaway for a chess engine builder? Is this a way to outperform stockfish for the same compute? More interpretable? More human?"
>
> We clarify that **chess language models—including our approach—are not designed to outperform state-of-the-art engines** in terms of raw playing strength. Traditional engines will always dominate in competitive settings due to their explicit search mechanisms and superhuman evaluation functions. However, our work offers three distinct practical benefits:
>
> 1. **Human-like play**
> 2. **Interpretability**
> 3. **Style preservation**
>
> In summary, our contribution is not to the competitive engine landscape, but to the development of **interpretable, human-aligned chess AI** that preserves the richness of individual playing styles.

---

> > ### Author Response · Authors · 2025-11-27
> > **Answer to Reviewer Rgkk [5/5]**
> >
> > We hope these clarifications address the reviewer's concerns and demonstrate the focused contributions of our work to both compositional modeling and human-aligned chess AI. Therefore, **we kindly ask for a reconsideration of the score**. We are happy to provide further details if needed.

---

> > > ### Comment · Reviewer_Rgkk · 2025-11-27
> > >
> > > This very long response does not contain much substance and in many of the responses simply plays word games (_Weakness 2: Text Representation and Legality_  is the worst example). I'm also concerned about the misstatements regarding a major citation in response to reviewer `4LHp`, and the repeated references to the _ chess language modeling literature_ which appears to be only 2 papers. As such I've lost faith in the validity of this work and lowered my score accordingly.

---

> ### Author Response · Authors · 2025-12-03
> **Final Answer to Reviewer Rgkk [1/6]**
>
> > [Reviewer] "You made two claims that I highlighted, I'm asking you to explain how you arrived at them." (McIlroy-Young et al. training data and code availability)
>
> > [Reviewer] "This very long response does not contain much substance and in many of the responses simply plays word games  (Weakness 2: Text Representation and Legality is the worst example). I'm also concerned about the misstatements regarding a major citation in response to reviewer 4LHp, and the repeated references to the _ chess language modeling literature_ which appears to be only 2 papers. As such I've lost faith in the validity of this work and lowered my score accordingly."
>
> We write to explicitly address the reviewer's stated "loss of faith in the validity of this work" and the consequent lowering of the score. We treat this declaration with the utmost seriousness.
>
> ### On "Substance" and "Word Games"
>
> We regret if the reviewer perceived our previous responses as lacking substance. We respectfully submit that our engagement has been rigorously substantive. We addressed every concern by providing **concrete data, verifiable implementation details, training logs, code/paper references, and new experimental results, along with actual manuscript revisions during the rebuttal period, strictly avoiding deferred promises of future corrections**. Despite our commitment to absolute transparency, the reviewer's continued concerns regarding misstatements and scientific validity compel us to provide a comprehensive response.
> In the following section, we provide the detailed derivation of our claims regarding the McIlroy-Young et al. (2021) paper as requested. We reiterate that this relates to the model-based behavioral stylometry metric introduced in our paper, a secondary contribution for *single-expert evaluation purposes only*.
>
> We remain acutely aware of the value of the reviewers' and Area Chairs' time. We assure you that we are not playing games.
> Regarding the specific criticism of "Weakness 2: Text Representation and Legality," we respectfully reject the characterization of our response as evasive.
> The reviewer's question—"why is a chess model getting less than 100% accuracy on picking a legal move OK?"—suggests a fundamental misalignment with the standard operating principles of autoregressive language models. Unlike symbolic engines, which are constrained by hard-coded rules, standard language models operate on probabilistic token prediction. Consequently, **the inability to guarantee 100% legality is an inherent architectural characteristic of the paradigm, not a methodological oversight or a flaw in our specific implementation, that actually improved the previous state-of-the-art seed models**. Our response aimed to clarify this distinction, which is well-established in the literature. We respectfully invite the reviewer to specify any arguments they perceive as *"word games,"* so that we may provide the necessary technical justification.

---

> ### Author Response · Authors · 2025-12-03
> **Final Answer to Reviewer Rgkk [2/6]**
>
> ### Clarification on McIlroy-Young et al. (2021)
>
> To remove all ambiguity, we provide the following technical summary of McIlroy-Young et al. (2021).
>
> **Model**
>
> - Individual moves (2x 34-channel 8x8 representation, before and after the player's move) $\rightarrow$ move features (a series of residual CNN blocks producing in output a 320-dim feature vector, then followed by a two-layer MLP to project it in a 1024-dim vector).
> - Move features $\rightarrow$ game features. Move features are processed by a ViT-based encoder with sinusoidal positional encoding. The 1024-dim output move vectors are averaged and projected to a 512-dim game vector with an additional MLP layer.
>
> **Training Loss**
>
> Game vectors trained by adapting the GE2E loss to maximize intra-player compactness and inter-player discrepancy.
> The centroid vector of each player is found by averaging all of their game vectors in the batch.
>
> **Dataset and Task Formalization: Training vs Reference vs Query Games**
>
> Players are bucked according to the number of games they have played: 1K–5K, 5K–10K, 10K–20K, 20K–30K, 30K–40K, and 40K+.
> Players are further divided into seen and unseen sets: seen if their games are used for model training, and unseen if not.
> Each player's games are randomly split into training games (80%), reference games (10%), and query games (10%).
> - Train games are available only for seen players and are used to train the move/game-encoder.
> - Reference games represent what is known about each candidate player.
> - Query games are what is given about an unknown target player we attempt to identify.
> So, the authors consider the following setting: the user is given a set of query games played by a particular player, and the task is to identify them from amongst a pool of candidate players, each of whom is associated with a set of reference games.
> In other words, the players we want to annotate the query games with are guaranteed to be members of a set of candidate players that may be equal to or greater than that set.
>
> **Predicted Identity = Top-1 Prediction**
>
> After training, the authors:
> 1. infer the player vector of each candidate player using the reference games;
> 2. infer the player vector of the unknown player using the query games;
> 3. compute cosine distance between the unknown player vector and all the candidate player vectors;
> 4. return the closest candidate as predicted identity.
>
> **Training Data (i.e., seen players and games)**
>
> - Lichess. 63.7M games by 16,181 amateur players, covering *all* buckets.
> - 9.5M games from 1,813 players coming from the top-1,500 leaderboard of both Lichess and Chess.com, having >= 950 played games. $\rightarrow$ *Only used in the Appendix.*
>
> **Evaluations in the Main Paper**
>
> In *almost* all their evaluations—excluding ablation studies, the authors use 100 games per player for both the reference and query sets.
>
> 1. Amateur players from Lichess.
> 	- Pool of candidate players (number of classes), C = 2,844 players (2,266 SEEN + 578 UNSEEN) with 10K-40K+ games played, i.e., 10K–20K U 20K–30K U 30K–40K U 40K+.
> 	- Query players, E:
> 		- (i) 578 UNSEEN Lichess players with 10K-40K+ (full generalization).
> 		- (ii) 2,266 SEEN (to check if it is easier to recognize players encountered during training).
> 2. High-ranked online players from Lichess and Chess.com (out-of-distribution examples).$^1$
>     - *"These players are not only unseen by the model, but they are of much higher skill than any of the players it was trained on, and half of them are drawn from a separate online platform".*
> 	- Pool of candidate players, C.
> 		- (i) ONLY UNSEEN. 400 out of the 408 UNSEEN high-ranked players, since used even in a previous work for comparison.
> 		- (ii) All the collected high-ranked players.
> 		- (iii) All the collected high-ranked players [+] SEEN (2,266) + UNSEEN (576) amateur players with 10K-40K+ games played used for (1).
> 	- Query players, E
> 		- The UNSEEN (400 or 408) high-ranked players.
>
> $^1$ Minor note. The paper is ambiguous on this part. In Table 1, the authors mention 451 unseen high-ranked players, while they say 408 in the main text. Similarly, in Table 1, the size of the complete set of high-ranked players is 2,264, while in the results table is listed as 2,157.
>
> **Relevant Appendix Results**
>
> - Accuracy in evaluating UNSEEN Lichess players based on the number of games played within the range 10K-40K.
> - Accuracy in evaluating SEEN+UNSEEN Lichess players when considering players belonging to 1K–5K, 5K–10K buckets as well.
> - Training new models on high-ranked players only.

---

> ### Author Response · Authors · 2025-12-03
> **Final Answer to Reviewer Rgkk [3/6]**
>
> ### McIlroy-Young et al. (2021) vs Ours (1/4)
>
> We now have a shared common knowledge to discuss the differences we claimed between McIlroy-Young et al. (2021) and our work.
>
> #### Goal, Stylometry Method and Experimental Setup
>
> > [Authors] "McIlroy-Young: 16K amateur players (Elo 1,000-2,000) + only 1.8K high-ranked players. Our work: 10 GMs exclusively, average Elo 2816, 418 points higher than their high-ranked subset. McIlroy-Young trained on 10K-40K games per player; we achieve comparable stylometric performance with only 1,000 games through vision foundation models."
>
> > [Reviewer] "I've familiar with McIlroy-Young et al. (2021) I don't recall it saying they trained on 10K-40K games per player"
>
> > [Authors] "Regarding dataset size and the reference to higher game counts, we specifically refer to Table 4 of McIlroy-Young et al. (2021). In this analysis, the authors stratify results by the number of games played, with categories extending up to 40K+. Their findings show that the embedding model’s performance improves substantially as the available per-player games increase. Our comparison was intended to emphasize data efficiency: whereas their method leverages large corpora to maximize accuracy, our vision-based approach is designed to identify elite Grandmasters using only 1,000 games per player. The contrast illustrates a difference in methodological regimes, not an implication that their work requires 10K+ games per player in evaluation."
>
> > [Reviewer] "You cite a table in the appendix [...] To be clear the McIlroy-Young et al. (2021) paper uses 100+100 (query + reference) games per player in all experiments, it compares to a previous benchmark that required 10k-40k which is discussed in both the main text and appendix."
>
> **McIlroy-Young et al. (2021)**
>
> - *Paper core contribution:* a **ViT-based encoder** model trained on millions of games from thousands of players to identify, at inference time, a player from among thousands of candidate players **given 100 labeled games**.
> - *Input board representation:* **feature-engineered**.
> - *Training data (population):* **63.7M games by 16,181 amateur players** for the main model.
> 	- The number of seen training games per player depends on the bucket, but it *includes* 10K-20K, 20K-30K, 30K-40K, and 40K+ cases; the average number of games per player across buckets is ∼4K. Even if the exact data distribution is not known and filtering operations may have been applied, approximate statistics can be evidenced by Table 1; e.g., 19 players x ∼40K games = ∼0.7M seen instances within the 40K+ bucket.
>     - *Hardware requirements:* 4x Tesla K80 GPUs, 24GB VRAM each.
> - *Evaluation:* **Precision@1; few-shot on seen and unseen players**, i.e., candidate player centroids initialized with the pooling of 100 query game embeddings.
>
> **Ours**
>
> - *Paper core contribution:* the first chess MoE model with small-sized GPT experts emulating world-class GMs.
>     - *What about behavioral stylometry then?* It serves here as a validation metric for expert fidelity, moving beyond strict memorization or exact-match ply accuracy. Unlike prior work focused on identifying a player from hundreds or thousands of candidates, our objective is verification: **determining whether the games generated by a specific expert align with the stylistic cluster of the target GM**. To achieve robust discrimination in this data-poor, elite regime, we fine-tune a **vision foundation model on raw board sequences augmented with hard visual prompting**.
> - *Input board representation:* **raw pixels; no feature-engineering**.
> - *Loss function:* while we inherit the core loss definition from Li Wan et al. (2018), further employed by McIlroy-Young et al. (2021), our loss **introduces two regularization terms** to address the increased complexity of our data setting, aimed to enforce inter-player separation and promoting intra-player compactness.
> - *Training data (population):* **10K games by 10 Super-GM players (2,816 Elo avg)**, 1000 games each, for the sake of White/Black bias removal.
>     - *Hardware requirements:* 1x NVIDIA GeForce RTX5090 GPU (32GB VRAM).
> - *Evaluation:* **Precision@K; zero-shot on seen players**, i.e., player embeddings (centroids) initialized with the pooling of training game embeddings.

---

> ### Author Response · Authors · 2025-12-03
> **Final Answer to Reviewer Rgkk [4/6]**
>
> ### McIlroy-Young et al. (2021) vs Ours (2/4)
>
> **Why is our setting difficult and unprecedented?**
>
> The inherent difficulty of our experimental regime—and its distinction from prior art—is empirically demonstrated by the McIlroy-Young et al. (2021) Appendix.
>
> - **Data volume.** The high-performance results reported in the main text of McIlroy-Young et al. (2021) focus on identifying amateur players with 10K or more games. Appendices 7.3 and 7.4 (Tables 4 and 5) reveal that expanding the evaluation pool to include players with fewer games significantly reduces identification accuracy. Specifically, **when the candidate and evaluation pools are broadened to encompass players with history depths of $\geq$ 1K games, P@1 degrades sharply from 0.860 to 0.540. Crucially, this decline occurs merely by *adding* lower-volume players to the high-data pool (10K–40K+), rather than restricting the evaluation exclusively to the low-data regime**. This provides the methodological rationale for the authors' exclusion of the 1K-5K and 5K-10K buckets from their primary analysis.
> - **Failure in the elite, small-cohort regime.**  In Appendices 7.7 and 7.8, the authors attempt a setting more aligned with our work: **training solely on a constrained cohort of high-ranked players**. They do this by alternatively using the 400 players from the previous work mentioned in the clarification section of our reply and the 1,813 players autonomously collected from Lichess and Chess.com. By quoting the authors, *"In the main text, we train our model on a large set of amateurs on Lichess. Here, we report on two additional experiments in which we train on a separate set of interesting players. In the first, we train our model only on the 400 players used in [31]. This is an experiment with training on orders of magnitude fewer players [...] In the second, we train on strong, high-ranking players from both Lichess and Chess.com, the two largest online chess platforms. This is also a smaller training set, and helps us understand whether our training methodology generalizes to high-level players."* We can focus on the first experiment (**N=400 high-ranked players**). Despite this cohort being 40$\times$ larger than our 10-GM set—a condition that theoretically facilitates more robust manifold learning—the model fails to retain discriminative power. Even when **evaluated on seen players (paralleling our evaluation protocol)**, **accuracy collapses to 0.080 (at $k=15$, mid-game start) and 0.243 (at $k=0$, game start)**. The authors explicitly attribute this failure to data scarcity: *"both models performed worse on their respective tasks than our main model due to the small training set size, and are included here for comparison and completeness. [...] **It does not generalize well and performs badly on the high-ranked players task**"*, and concluding that ***more than 400 players are needed to learn the space of chess-playing style**.*.
> - **More accentuated stylistic overlap.** These $k$-sensitive results not only highlight the complexity of our case but also the **difference in data distributions** between our paper and McIlroy-Young et al. (2021). In the high-ranked games considered by the latter, the opening choice appears to be highly discriminative, leading to a performance delta of ∼20 percentage points. Instead, as we analyzed in detail in Appendix C.1, in our dataset, each player's opening repertoire is generally dominated by two or three primary systems, with preferences often overlapping or similar; stylometric cues are considerably subtler. Consequently, unlike their dataset where openings serve as a strong "tell," **our regime requires detecting considerably subtler stylometric cues in the middlegame and endgame**, confirming that **the two studies operate on statistically distinct populations**.
> - **Summary.** Verifying the stylistic fidelity of Super-GM (Avg Elo 2,816) emulators under a strict 1,000-game constraint constitutes a substantially more complex task than amateur identification. This complexity arises from the intersection of extreme data scarcity and high distributional overlap (driven by the objective optimality of elite play). **To our knowledge, this specific elite, data-poor regime has not been targeted in the literature, and we respectfully submit that the magnitude of this challenge was underestimated in the assessment, including by Reviewer HMfV**.

---

> ### Author Response · Authors · 2025-12-03
> **Final Answer to Reviewer Rgkk [5/6]**
>
> ### McIlroy-Young et al. (2021) vs Ours (3/4)
>
> #### Code Availability
>
> > [Authors] "Furthermore, the codebase and model weights for McIlroy-Young et al. (2021) are well-known to be not publicly available due to ethical concerns, with key implementation details not derivable from the paper. Consequently, establishing a fair baseline would require a complete re-implementation with degrees of freedom and re-training on massive datasets to which we do not have access for reproducibility before adaptation."
>
> > [Reviewer] "I've familiar with McIlroy-Young et al. (2021). I don't recall it saying that the weights are unavailable. Can you double check that's what the paper says?  --> Our public code release can be found at github.com/CSSLab/behavioral-stylometry. As there are ethical concerns regarding releasing the model and full training code it may not be publicly accessible. You can contact our corresponding author ... to request access."
>
> > [Authors] "With respect to reproducibility and baseline comparison, we requested access to the code and resources early in the project, but did not receive a response. Given the time since publication and the ethical considerations surrounding the project, we did not continue pursuing it. Consequently, we developed an independent system with distinct architectural, theoretical, and perceptual components. Without the original code or pretrained weights—and given the substantial methodological differences (symbolic feature engineering vs. raw-video vision modeling)—a faithful reproduction was not feasible."
>
> > [Reviewer] "You do not explain your second statement just that you gave up after a single attempt."
>
> As noted in our initial response and correctly identified by the reviewer, **the codebase and model weights for McIlroy-Young et al. (2021) have never been publicly released due to stated ethical concerns; a status verifiable via the Internet Archive**. Following the authors' instructions, we contacted the corresponding author at an early stage of this project to request access. Unfortunately, **we received no response**; we can document our statement. Given the lack of a reply and the elapsed time since the paper's publication (over four years), we reasonably concluded that the repository might no longer be actively maintained.
> **We also tried to re-implement the work from scratch by ourselves**, with the goal of applying the proposed architecture to our case. However, the preliminary **results were poor and not encouraging**; moreover, we had to make custom implementation choices for details not clearly specified in the paper, which may have deviated from the original work. Given the insufficient identification performance and uncertainty regarding 1:1 fidelity, we decided to discard this work line rather than present a potentially flawed comparison that would misrepresent the quality of McIlroy-Young et al. (2021).
> In light of these reproducibility challenges—and considering our **need to achieve a sufficiently good proxy metric for player emulation with a lightweight solution targeting a very small number of GMs**—we moved to a different, original, and modern methodological approach. Motivated by recent milestones in visual representation learning (zero-shot/few-shot capabilities) and qualitative feedback from our survey of experienced players regarding the cognitive salience of visual patterns, **we pivoted to pioneering the specialization of pre-trained vision encoders for chess behavioral stylometry**.
> Throughout this process, we have maintained the utmost respect for the foundational work of McIlroy-Young et al. (2021), properly citing it six times, together with the other relevant papers from the same authors, in what—to our knowledge—is one of the most extensive Related Work analysis in the chess AI literature.
> **Should the reviewer be able to facilitate access to the original McIlroy-Young et al. (2021) models, we would welcome the opportunity to include a head-to-head performance comparison with our vision-based encoder. However, based on the distributional shifts and data constraints detailed previously, we maintain a strong scientific conviction that the baseline model would require significant ad-hoc retraining to function as a reliable validation metric for the experts in our primary contribution, the Mixture-of-Masters architecture—a contribution that we feel remains under-recognized in the current assessment.**
> Ultimately, we respectfully submit that this rigorous process of outreach, attempted re-implementation, and strategic methodological pivot cannot be accurately characterized as *"giving up after a single attempt."*

---

> ### Author Response · Authors · 2025-12-03
> **Final Answer to Reviewer Rgkk [6/6]**
>
> ### McIlroy-Young et al. (2021) vs Ours (4/4)
>
> #### Model Type
>
> We fully recognize McIlroy-Young et al. (2021) as an embedding model used for classification tasks.
> This understanding is reflected in our previous statements:
>
> > [Authors] "Their {McIlroy-Young et al. (2021)} findings show that the embedding model’s performance improves substantially as the available per-player games increase."
>
> > [Authors] "Player classification as primary goal"
>
> > [Authors] "{We consider} top-k retrieval, rather than claiming strict diagonal dominance. [...] This retrieval-based evaluation is appropriate given the subtle divergences characteristic of elite-level play."
>
> We used the term "classifier" functionally to distinguish it from our retrieval-oriented evaluation protocol.
> We regret if our previous characterization of their candidate-based evaluation as "closed-set" appeared as an oversimplification of their architecture's capabilities.
> Specifically, **we consider the Top-$k$ nearest centroids, diverging from the strict Top-1 identification metric relied upon by McIlroy-Young et al. (2021). In the context of elite play, a Top-1 metric would fundamentally mischaracterize the nature of chess style: as validated by our expert survey and distributional statistics reported in Appendix, the GMs targeted by our study share a substantial base of objective strategic knowledge, leading to inevitable repertoire overlaps.** Consequently, valid emulation often maps to a stylistic neighborhood rather than a unique, disjoint identity. **Althoguh we acknowledge that the architecture proposed by McIlroy-Young et al. is theoretically capable of supporting such retrieval tasks, their original analysis does not explore it.**
>
> ### Literature
>
> We respectfully correct the Reviewer's misconception that the "chess language modeling literature" consists of only two papers. As detailed in our Related Work, this is a rapidly expanding subfield. **Our design choices were informed by 30+ distinct studies**, which we summarize to contextualize our contributions. **Although McIlroy-Young et al. (2021) is a key reference for stylometry, it is the broader language modeling literature that primarily drives our core contribution: the Mixture-of-Masters architecture**.
>
> ### Overall Assessment: from 6 to 0
>
> We respectfully but firmly reject the accusation of "misstatements" and the stated "loss of faith" in the scientific validity of our work.
>
> We have approached the submission and rebuttal process with maximum diligence, providing:
> - Concrete Evidence: Verifiable training logs, code references, and mathematical derivations.
> - Responsiveness: New experiments and manuscript revisions executed during the rebuttal window.
> - Transparency: A commitment to Open Science, evidenced by our anonymous repository, playable demos, and datasets.
>
> These are the actions of a research team dedicated to rigor, not one engaging in "word games."
>
> We find the reduction of the score from 6 (Weak Accept) to 0 (Strong Reject) to be objectively unjustified and disproportionate to the concerns raised and their resolution. A "Strong Reject" at ICLR is reserved for submissions that are critically flawed, specifically:
> - Papers lacking any scientific merit or containing pervasive technical errors.
> - Papers with fabricated data, plagiarism, or severe ethical violations.
> - Papers that fail to offer any new knowledge to the community.
>
> We strongly believe that our submission does not meet any of these criteria. To assign the lowest possible score based primarily on a questionable disagreement regarding the details of a secondary evaluation metric, while ignoring the primary language modeling contributions, constitutes a failure to evaluate the paper on its holistic scientific merit. The reviewer’s critique has narrowed almost exclusively to the McIlroy-Young et al. (2021) citation, on which we reiterated.
>
> **The reviewer's engagement—characterized by short, punitive responses, accusations of dishonesty without evidence, and a refusal to acknowledge all the contributions of our paper and the extensive substantive rebuttals provided—does not appear to align with the constructive and fair principles of the ICLR Code of Ethics.**
>
> A score reduction of this magnitude, predicated on a single citation dispute and a subjective **"loss of faith" despite reproducible evidence to the contrary**, is scientifically unsound. We kindly request that the Area Chair and other reviewers assess the validity of our work based on all the contributions and the empirical evidence provided, rather than this singular, disproportionate point of contention.

---

### Official Review · Reviewer_8mhw · 2025-11-01

**Soundness:** 3
**Presentation:** 4
**Contribution:** 4
**Rating:** 8
**Confidence:** 4

**Summary:**

The paper introduces Mixture of Masters (MOM), a sparse chess language model that combines multiple small GPT-based experts, each trained to emulate the distinctive style of a world-class grandmaster. A learnable gating network dynamically selects which expert to consult for each move, enabling the model to switch between offensive, positional, or defensive styles depending on the game state.
MOM addresses the homogenization problem of dense chess models that lose stylistic diversity. The authors also propose a behavioral stylometry metric using a vision transformer to verify that each expert maintains a unique playing signature. Experiments show that MOM preserves creative variability and outperforms both dense GPT baselines and single-expert models when evaluated against Stockfish on unseen games.

**Strengths:**

- This work introduces a very insightful framework, which is the first persona-based sparse MoE chess language model. Each small GPT expert emulates a specific grandmaster and dynamically routes by game state. I can see similar ideas easily be extended to other domains where individual styles or strategic personas matter.
- The experiments present extensive ablations and Stockfish evaluations under strict legality rules, demonstrating consistent improvements and interpretable routing behavior across different difficulty levels.
- The content of this paper is well organized and easy to follow.

**Weaknesses:**

- My main concern is that the scalability and efficiency of the proposed method are not deeply analyzed. Training and maintaining multiple experts plus a gating network could be computationally expensive compared with a single dense model, limiting its practicality for larger expert sets.
- The acronym MOM is a bit off.

**Questions:**

- How exactly is the gating network trained? Does it receive supervision from expert identities, or only learn through downstream loss during move prediction?
- When merging weights across experts, which layers are merged and which are gated, and how is alignment between different expert representations ensured?
- Is it possible for MOM to generalize to unseen players or mixed-style games?

---

> ### Author Response · Authors · 2025-11-27
> **Answer to Reviewer 8mhw [1/2]**
>
> We sincerely thank the reviewer for the thoughtful and encouraging evaluation. We appreciate the careful reading and constructive questions—these comments helped us refine the clarity of our method and better highlight its computational properties and generalization behavior. We address each point below.
>
> ### **Scalability and Computational Efficiency**
>
> We agree that the scalability of multi-expert systems is an important consideration. In MOM, the computational overhead remains modest for three key reasons:
>
> 1. **Only a subset of layers is parallelized.**
>    As detailed in §3.2, we gate only the attention projections (Q/K/V and output-projection layers), which represent a limited fraction of the overall transformer compute. The remaining layers are merged into a shared backbone and executed once.
> 2. **We activate only *k = 2* experts at inference.**
>    Our ablation in Figure 7 shows that *k = 2* yields the best performance/efficiency trade-off. This means that, even with multiple experts available, MOM parallelizes computation across just two branches per routed layer.
> 3. **Experts are lightweight (50M parameters each).**
>    Each expert is built from the nanoGPT family, making them significantly smaller than typical MoE experts in LLM literature.
>
> All routed computations are parallelized on GPUs, and training/inference wall-clock time remains close to that of a single-expert model. However, to provide clearer and more precise insights, we are currently analyzing latency for the baseline solution and various MoM configurations. Due to time constraints, we will include these statistics in the next phases.
>
> This demonstrates that MOM is not only conceptually scalable but also practically tractable for larger expert sets.
>
> ### **Training of the Gating Network**
> > [Reviewer] "How exactly is the gating network trained? Does it receive supervision from expert identities, or only learn through downstream loss during move prediction?"
> The gating network is trained **fully end-to-end** with the downstream next-move prediction objective. No supervision from expert identities is used during routing.
>
> - During the *stitching* phase, we discard all information about which expert generated which game.
> - The gating network receives only board-state embeddings and learns the optimal expert combination *implicitly* through backpropagation from the move-prediction loss.
> - This follows the standard MoE practice described in §3.2, using differentiable Gumbel-Softmax for top-k routing during training.
>
> Thus, MOM autonomously discovers when and how to use each GM expert.
>
> ### **Weight Merging, Gated Layers, and Alignment**
> > [Reviewer] "When merging weights across experts, which layers are merged and which are gated, and how is alignment ensured?"
> As described in §3.2, our stitching procedure follows a hybrid pattern in which only a subset of layers remains expert-specific. The attention projections (Q/K/V and the attention output projection) are kept separate for each expert and routed through the gating network, whereas the rest of the transformer stack—including embeddings, attention heads, feed-forward layers, and positional encodings—is merged into a shared backbone using uniform averaging. This design preserves the stylistic and behavioral specialization within the attention mechanisms while consolidating the broader representational pipeline into a single coherent model.
>
> To ensure that these merged components interact smoothly, we introduce a brief alignment fine-tuning phase where the shared parameters and the router are jointly trained on a mixture of seed-model and GM-specific data (§4.1). This step empirically stabilizes the shared representation space and prevents interference across experts. Additional implementation details are provided in Appendix D.
>
> ### **Generalization to Unseen Players and Mixed-Style Games**
>
> > [Reviewer] "Is it possible for MOM to generalize to unseen players or mixed-style games?"
> MOM is specifically designed to generalize beyond the personas used during expert pretraining. Because the router is trained without any supervision from player identities, it learns to base its routing decisions on the structure of the position itself. As a result, when presented with moves from an unseen player, MOM naturally allocates different phases of the game to the experts whose styles best match the evolving board dynamics.
>
> This mechanism also enables the model to handle mixed-style or phase-shifting games. As illustrated in Figure 9, the routing pattern adapts continuously over the course of a single match, allowing the model to follow shifts between opening theory, tactical middlegame play, and positional or technical endgames. Furthermore, MOM performs strongly against Stockfish, whose play does not resemble any individual expert, demonstrating robustness to stylistically unfamiliar inputs. We have added a brief clarification in the revision to make this generalization behavior more explicit.

---

> > ### Author Response · Authors · 2025-11-27
> > **Answer to Reviewer 8mhw [2/2]**
> >
> > We thank the reviewer again for the positive assessment and for highlighting important points of clarification. MOM was intentionally built as a lightweight, modular, and computationally efficient MoE architecture, and we appreciate the chance to better articulate these strengths. We hope that the revised explanations make the design choices and their practical implications clearer.

---

### Official Review · Reviewer_4LHp · 2025-11-04

**Soundness:** 3
**Presentation:** 2
**Contribution:** 2
**Rating:** 2
**Confidence:** 4

**Summary:**

This paper proposes mixture of masters, a technique to counteract homogeneity in chess models.

**Strengths:**

* The paper is well written.
* The illustrations and presentation are clear.

**Weaknesses:**

* The paper claims to contribute "behavioral stylometry" for chess, but this was first done in McIlroy-Young et al 2021 (not cited in the behavioral stylometry section). The method is even very similar, and the paper sounds like it is claiming it for itself (e.g. "Drawing inspiration from speaker recognition systems, we employ the generalized end-to-end (G2E2) loss", which is exactly what McIlroy-Young et al 2021 do). The ethics statement also makes it sound like the behavioral stylometry metric is an original contribution "Our introduction of a behavioral stylometry metric...".

* Despite the substantial overlap with previous work, it is not used as a baseline in the experiments, so it's difficult to know how it compares.

* In Figure 8b, the position is illegal, since it is White to move but Black is in check. If simple errors like this example are already visible in the figures, this makes me seriously doubt the validity of the paper's results.

**Questions:**

How does this method differ from previous work?

How do other baselines do?

**Details Of Ethics Concerns:**

See weaknesses #1. There is substantial unacknowledged overlap with previous work.

---

> ### Author Response · Authors · 2025-11-26
> **Answer to Reviewer 4LHp [1/5]**
>
> We thank the reviewer for the comments provided.
>
> ### 1. Originality (Weakness 1, Question 1)
>
> > [Reviewer] "The paper claims to contribute "behavioral stylometry" for chess, but this was first done in McIlroy-Young et al 2021 (not cited in the behavioral stylometry section). The method is even very similar, and the paper sounds like it is claiming it for itself (e.g. "Drawing inspiration from speaker recognition systems, we employ the generalized end-to-end (G2E2) loss", which is exactly what McIlroy-Young et al 2021 do). The ethics statement also makes it sound like the behavioral stylometry metric is an original contribution "Our introduction of a behavioral stylometry metric..."."
>
> **The claim that McIlroy-Young et al. 2021 is "not cited in the behavioral stylometry section" is factually incorrect.** We cite this work extensively throughout the paper, including prominent discussions within the Related Work and Behavioral Stylometry sections, where we explicitly contrast our methodologies.
>
> > [Authors, Related Work, L86-90] "Finally, we contribute a novel form of behavioral stylometry; while existing models for player identification operate on symbolic move data with human-engineered features (McIlroy-Young et al., 2021; 2022), we operate on raw game video recordings directly."
>
> > [Authors, Extended Related Work, L1390-1411] "Closer to our work, the same authors proposed models capable of identifying hundreds of individual players from their move patterns: first by fine-tuning Maia-1900 (McIlroy-Young et al., 2022), then by training a vision transformer from scratch on sequences of moves represented as 3D tensors with human-engineered features (McIlroy-Young et al., 2021). Influenced by this research line, we create individual experts and implement a model-based behavioral stylometry metric, eliminating the need for feature engineering by operating directly on \textit{raw game video recordings}.
> For context, McIlroy-Young et al. (2021) trained on 10K to 40K games from 16K players of Elo 1,000-2,000 and only 1.8K high-ranked players selected from lichess and chess.com leaderboards. Instead, we target 10 GMs with an average 2,816 Elo, and train vision foundation models to obtain appreciable performance with few training data, 1,000 games."
>
> **These citations appear in the exact sections the reviewer claims they are missing.** The statement "Drawing inspiration from speaker recognition systems" is accurate, as GE2E loss was originally proposed for speaker verification (Wan et al., 2018), and both our work and McIlroy-Young et al. adapt it to chess.
>
> Moreover, we cite McIlroy-Young et al. 2021 in two additional locations:
>
> > [Authors, RQ3 Discussion, L405-407] "Identifying stylistic patterns among elite chess players remains a challenging problem, as classification-based approaches have achieved limited accuracy (McIlroy-Young et al., 2021)."
>
> > [Authors, Player Recognizability, L974-977] "Some experts argue that professional players can indeed be recognized from their moves alone, pointing to recent machine learning studies that achieve high accuracy in attributing games even when results and openings are excluded McIlroy-Young et al. (2021), suggesting that mid- and late-game decisions carry individual traces."

---

> ### Author Response · Authors · 2025-11-26
> **Answer to Reviewer 4LHp [2/5]**
>
> ### 1. (Continue)
>
> **Clarification on contribution scope:** We explicitly reject the interpretation that we claim to have invented the concept of behavioral stylometry. We acknowledge McIlroy-Young et al. (2021) as the pioneers of this research direction throughout our manuscript. However, our contribution represents a **fundamental paradigm shift** designed to resolve specific, critical limitations in their approach.
> We distinguish our methodological contributions from their work on four decisive axes:
> - *Modality (feature engineering vs. representation learning):* Their approach relies on training vision transformers from scratch using symbolic 3D tensors and human-engineered features. We introduce a vision-based paradigm leveraging pretrained foundation models (DINOv3) on raw game video, eliminating feature engineering entirely.
> - *Data Efficiency:* Their method demands massive datasets (10k–40k games per player). We achieve comparable stylometry performance with one order of magnitude less data (1,000 games per player).
> - *Extensibility (Classification vs. Retrieval):* Their classification-based framework necessitates full model retraining to add new players. We employ a more flexible retrieval-based framework that utilizes cosine similarity between game embeddings and player centroids. This allows for open-set recognition where new players are added by simply computing a centroid, providing continuous similarity scores rather than discrete class labels.
> - *Target Population:* Their work documents a severe performance degradation at the elite level (a 64% accuracy drop from amateur to high-ranked players, Table 3, McIlroy-Young et al., 2021). Our method successfully targets the elite GM population (avg. Elo 2,816), a segment where their approach proved ineffective.
>
> Therefore, when the ethics statement refers to "our behavioral stylometry metric," it addresses the specific implications of this efficient, vision-based, retrieval-oriented tool—particularly its ability to fingerprint players with minimal data—rather than the general concept of chess stylometry, which remains fully attributed to prior work.
>
> Finally, we find the reviewer's implication regarding research integrity—specifically the suggestion that we attempted to misappropriate the general concept of behavioral stylometry—particularly concerning and objectively unfounded. We urge a re-examination of these sections, as the allegation of missing citations and misattribution is factually contradicted by the manuscript's text.

---

> ### Author Response · Authors · 2025-11-26
> **Answer to Reviewer 4LHp [3/5]**
>
> ### 2. Stylometry Baselines (Weakness 2, Question 2)
>
> > [Reviewer] "Despite the substantial overlap with previous work, it is not used as a baseline in the experiments, so it's difficult to know how it compares."
>
> The reviewer's request for baseline comparisons misunderstands the fundamental differences in our research objectives, methodologies, and target populations. McIlroy-Young et al. (2021) explicitly acknowledge that their approach faces substantial challenges when applied to elite players, which is precisely the population we target.
>
> **Evidence from McIlroy-Young et al. (2021):**
>
> Their own results demonstrate that stylometry accuracy **drops by 64%** when moving from amateur to elite players (Page 8, Table 3):
> - Amateur players (Elo 1,000-2,000): P@1 = 0.860
> - High-ranked players: P@1 = 0.308
>
> They explicitly state: *"This task is significantly more difficult: both the baseline and our model have lower P@1 scores"* (Page 8). Furthermore, they characterize elite players as out-of-distribution: *"Even when trained on amateur play, our method generalises to out-of-distribution samples of Grandmaster players, despite the dramatic differences between amateur and world-class players"* (Page 1, Abstract).
>
> **Critical differences that preclude direct comparison:**
>
> 1. **Target Population & Data Scale:**
>    - McIlroy-Young: 16K amateur players (Elo 1,000-2,000) + only 1.8K high-ranked players
>    - Our work: 10 GMs exclusively, average Elo 2816, **418 points higher** than their high-ranked subset
>    - McIlroy-Young trained on 10K-40K games per player; we achieve comparable stylometric performance with only 1,000 games through vision foundation models
>
> 2. **Input Representation:**
>    - McIlroy-Young: Symbolic 3D tensors with human-engineered features
>    - Our work: Raw game video with pretrained vision transformers (DINOv3), eliminating feature engineering
>
> 3. **Objective:**
>    - McIlroy-Young: Player classification as primary goal
>    - Our work: Stylometry as a **validation metric for expert specialization** within a mixture-of-experts architecture for move generation
>
> 4. **Evaluation Protocol:**
>    - McIlroy-Young: Classification accuracy on hundreds/thousands of players
>    - Our work: Cosine similarity between game embeddings and GM centroids for 10 experts, integrated within MoE move prediction
>
> The McIlroy-Young stylometry model is not a suitable baseline because it addresses a different task (classification vs. expert validation), uses a different methodology (symbolic engineered features vs. vision-based), and targets a different population (amateurs vs. elite GMs)). Furthermore, the codebase and model weights for McIlroy-Young et al. (2021) are well-known to be not publicly available due to ethical concerns, with key implementation details not derivable from the paper. Consequently, establishing a fair baseline would require a complete re-implementation with degrees of freedom and re-training on massive datasets to which we do not have access for reproducibility before adaptation.

---

> > ### Comment · Reviewer_Rgkk · 2025-11-26
> >
> > I've familiar with _McIlroy-Young et al. (2021)_ I don't recall it saying they trained on 10K-40K games per player, Or that the weights are unavailable. Can you double check that's what the paper says?
> >
> > Because I see on the ArXiv version:
> >
> > > bucket the players according to how many games they have played: 1K–5K, 5K–10K, 10K–20K, 20K–30K, 30K–40K, and 40K+
> >
> > And
> >
> > > Our public code release can be found at github.com/CSSLab/behavioral-stylometry. As there are ethical concerns regarding releasing the model and full training code it may not be publicly accessible. You can contact our corresponding author ... to request access.

---

> > > ### Author Response · Authors · 2025-11-27
> > > **Answer to Reviewer Rgkk on stylometry baselines**
> > >
> > > We thank the reviewer `Rgkk` for engaging with the discussion raised in the other reviews and for prompting a deeper clarification of how our contributions differ from prior work.
> > >
> > > We would like to restate our position regarding the comparison to McIlroy-Young et al. (2021) and explain in more detail why their experimental analysis is not aligned with the setting, constraints, or objectives of our study.
> > >
> > > Regarding dataset size and the reference to higher game counts, we specifically refer to **Table 4 of McIlroy-Young et al. (2021)**. In this analysis, the authors stratify results by the number of games played, with categories extending up to 40K+. Their findings show that the embedding model’s performance improves substantially as the available per-player games increase. Our comparison was intended to emphasize **data efficiency**: whereas their method leverages large corpora to maximize accuracy, our vision-based approach is designed to identify elite Grandmasters using only **1,000 games per player**. The contrast illustrates a difference in methodological regimes, not an implication that their work requires 10K+ games per player in evaluation.
> > >
> > > We also acknowledge that the authors include bucket analyses for players with a limited number of games. However, this setup does not examine performance under both **(i)** a low number of games per player **and** **(ii)** a very small, elite-only pool of players—precisely the scenario that defines our work. In addition, the ≈1K-game bucket is composed predominantly of **low-ELO players**, who are empirically easier to classify due to higher stylistic variance and simpler tactical patterns. Such characteristics fundamentally change the classification difficulty. In contrast, our method tackles the much more demanding problem of clustering **highly similar, elite-level Grandmasters**, where stylometric cues are considerably more subtle. Because of these population and difficulty mismatches, their bucket-level results cannot serve as a meaningful baseline for our setting.
> > >
> > > With respect to reproducibility and baseline comparison, we requested access to the code and resources early in the project, but did not receive a response. Given the time since publication and the ethical considerations surrounding the project, we did not continue pursuing it. Consequently, we developed an independent system with distinct architectural, theoretical, and perceptual components. Without the original code or pretrained weights—and given the substantial methodological differences (symbolic feature engineering vs. raw-video vision modeling)—a faithful reproduction was not feasible.
> > >
> > > Even if pretrained weights and code had been available, a direct numerical comparison would remain scientifically inappropriate due to key misalignments:
> > > * **Task Definition:** Their model is a **closed-set classifier**, assigning a game to one of \(N\) fixed identities. Our system performs **retrieval** using continuous embedding similarity, enabling open-set recognition and top-\(k\) confidence estimation—capabilities absent in their formulation.
> > > * **Target Population:** Their model was optimized for amateur and intermediate-level players. Applying it to elite Grandmasters introduces a significant distribution shift, requiring full retraining and extensive hyperparameter retuning.
> > >
> > > For these reasons, using McIlroy-Young et al. (2021) as a baseline would not yield a fair or interpretable comparison. That said, their foundational role in behavioral stylometry is clearly acknowledged throughout our manuscript, including in Related Work, the Behavioral Stylometry discussion, RQ3, and the Appendix. Our contribution lies in a distinct instantiation designed specifically for the elite-player regime, which carries substantially different constraints and objectives.
> > >
> > > We hope this provides full clarity on the methodological differences and the rationale behind our experimental choices.

---

> > > > ### Comment · Reviewer_Rgkk · 2025-11-27
> > > >
> > > > You made two claims that I highlighted, I'm asking you to explain how you arrived at them. You cite a table in the appendix and do not explain your second statement just that you gave up after a single attempt.
> > > >
> > > > To be clear the _McIlroy-Young et al. (2021)_ paper uses 100+100 (query + reference) games per player in all experiments, it compares to a previous benchmark that required 10k-40k which is discussed in both the main text and appendix. It also is not a classifier, it is an embedding model, it was used for classification, figure 3c shows highlights this.

---

> ### Author Response · Authors · 2025-11-26
> **Answer to Reviewer 4LHp [4/5]**
>
> ### 3. Qualitative Example (Weakness 3)
>
> > [Reviewer] "In Figure 8b, the position is illegal, since it is White to move but Black is in check. If simple errors like this example are already visible in the figures, this makes me seriously doubt the validity of the paper's results."
>
> As noted in our rebuttal to Reviewer HMfV, this error stemmed from a manual oversight during the derivation of the FEN string from the game PGN, a step required to render the board visualization in LaTeX using the xskak package. The fact that this is a simple visualization oversight is objectively verifiable via the PGN notation provided immediately above the board in the figure, which correctly lists the Black King's position at f8 rather than f7. We have rigorously verified the moves predicted by our MoM model and the corresponding expert activation patterns, confirming that the underlying data and the results reported in the manuscript remain fully accurate. Consequently, we have regenerated the board visualization using the correct FEN string in the revised manuscript.
>
> Furthermore, we find that the allegation that a LaTeX rendering error casts doubt on the validity of our entire study is disproportionate and disrespectful.
> It extrapolates a fundamental scientific flaw from a minor LaTeX rendering oversight, ignoring the extensive empirical evidence provided.
> Our work adheres to the highest standards of scientific transparency and is fully reproducible.
> Our complete experimental pipeline, including 6,000+ evaluation games with automated validation, is publicly available in the anonymous GitHub repository provided during the review process.
> We have deployed a playable bot on Lichess (https://lichess.org/@/mixture-of-masters), where reviewers and users can interact with our MoM models in real-time to verify its legal play and strategic capabilities firsthand.

---

> ### Author Response · Authors · 2025-11-26
> **Answer to Reviewer 4LHp [5/5]**
>
> We respectfully find the review superficial, characterized by significant misconceptions and a failure to recognize the full scope of the proposed contributions.
>
> We have provided evidence regarding: (1) the rigorous and extensive citation of McIlroy-Young et al. throughout the manuscript, refuting the claims of non-originality; (2) the fundamental methodological divergences that render a direct baseline comparison scientifically inappropriate; and (3) the validity of our approach, confirmed by performance on elite GMs—a domain where prior methods are known to suffer severe degradation.
>
> The concerns regarding originality and the associated ethics flag appear to stem from an incomplete reading of the manuscript, specifically overlooking the explicit citations and distinctions detailed in the Related Work and Methods sections. Furthermore, we reiterate that a single visualization artifact in a figure does not invalidate a rigorous, fully-transparent experimental protocol, a publicly released codebase, and playable models.
>
> We believe a comprehensive re-examination of the manuscript will reveal that this work represents a distinct advancement beyond prior art. We target critical challenges in interpretability, controllability, and style homogenization by introducing—for the first time—GM language model emulators (trained via SSL+RL), chess MoE architectures with player routing, and visual behavioral stylometry metrics devoid of manual feature engineering. We set new state-of-the-art performance following open-science principles.
> **In light of these clarifications, we request a re-evaluation of the score. We trust that the reviewer will recognize the extent of these contributions in alignment with the assessment of the other reviewers.**

---

### Official Review · Reviewer_viS1 · 2025-11-05

**Soundness:** 2
**Presentation:** 2
**Contribution:** 2
**Rating:** 4
**Confidence:** 3

**Summary:**

This paper proposes to split Chess Language Models into distinct experts each trained following a different real Grandmaster (GM), and using those expert models collectively in a Mixture-of-Experts architecture. It grounds this idea in the notion that chess player have individual styles, including strength and weaknesses, where a given style might be most suited to a specific game situation.
The paper also proposes a systematic analysis of the different expert styles through a behavioral stylometry.

**Strengths:**

The main idea of the paper, splitting move selection based on position to use a more appropriate model is interesting, has potential to improve current Chess Language Models, and provide for some improved, though still nebulous, method to interpret move decision.

This paper providing new tools to have stylistically different Chess Language Models is welcome, although the ethical issues that could arise (and are well mentioned by the authors in the second to last section of the main paper) should be kept in mind.

**Weaknesses:**

The whole stylometry axis of the paper is quite confused:
- The explanation of the methodology is close to unintelligible, due to the explanation being far too compressed and out of order, as well as a notational mess. For instance, $h_k$ is defined as the average of all k-th tokens over the temporal window [i, i+F-i] (which is only vaguely mentioned by text), and are then used to define $h_j$, which has no dependency on j (all $h_j$ are equal), unless maybe j here refers to i in $h_k$??? In that case, the $h_j$ would be a representation of position j, as well as all the F-1 following positions. The whole loss section also has troubles, that are explicited further in the next section.
- The results of the stylometry analysis are incomplete, and poorly analyzed. So far as I can tell, they are only reported in Table 3 (labeled stilometry), and explained in RQ3. The table doesn't really show that "each expert model is stylistically closest to its corresponding target master" (L410). Only half of the experts have highest similarity with their master, and only 2 (maybe 4) can be said to be explicitly closest (3 and 10, maybe 5 and 9). 7 and 8 are not even really among the top 3.
- An additional stylometry table comparing real games (unused in training data) from the 10 sampled masters would be really helpful, either in the main paper or in an appendix, to support the main table results, and to provide more context on how well their "styles" can be differentiated.

Overall, the report and presentation of the results of this paper feel confused. In addition to the issues with the stylometry results,
- In RQ2 looking into the impact of RL on individual experts, figure 5 is cited to justify a substantial increase in the ability to generalize and explore diverse options, when the figure only shows that the generated moves are slightly more likely to be legal (maybe around +2%?).
- Maybe the correct figure for this should be Table 2? But even then adding RL usually lowers win rate and fide score compared to using only SSL. The explanation of the results might be mentioning the full MoM model, but it doesn't mention that, is between RQ1 and RQ3, which both only look at experts individually, giving the impression that the first 3 questions focus on the experts, while the later 2 on the full MoM model.

**Questions:**

The stylometry loss definitions were quite hard to follow:
- $G_p$ is used both as a set and a number. $\frac{1}{G_q-1}$ should be $\frac{1}{|G_q|-1}$ in Eq (3) for instance. also, $\\{G_q\setminus{}g\\}$ should just be $G_q\setminus{}g$. Also in Eq (3), $v_g^q$ is used. Is is meant to be $z$ instead of $v$? Also in this equation, the contribution of $z_g^p$ is suppressed on its own centroid, but is done in a weird way. When suppressed, there is no issue. When not suppressed, the centroid becomes the sum of the previous centroid and $z_g^q$, not the average of all $z_\cdot^q$.
- $L_{MoM}$ is almost impossible to parse. n_p and G_p are not appropriate notations in an equation summing over p independently. In the second term of Eq (4), g and g̃ are used, but not quantified. Is is supposed to be a quadruple sum and not a double sum? In that case, why is that term not normalized by $|G|^2$? And overall, the choice of normalization is a bit cryptic. the first and last term are normalized by |G|, while the second term is normalized by $n^2$. Is there a reason not to just have each sum be a mean instead, keeping both lambdas?

- For Figure 5, the move legality seems to be on average 80 to 90. What is the unit? I assume those are percentages for now. I am confused on how the models are able to play with any consistency when once in every 10 moves they instantly lose the game.
- In Figure 6, the FIDEScore seem to hover around the high 60s, being maybe lower for k=5. However, in Figure 7, MoM seems to only score around 59 (percent? total score over 100 games maybe? Which would be the same in practice). Maybe the legend of Figure 6 meant to mention Stockfish level 0 instead of Stockfish 1?
- In Figure 8, the second image shows the black king on f8, when it should have moved (out of check!) 2 plys earlier.

As a very minor point, the introduction mentions that the number of chess games (~$10^{50}$ in the paper, currently estimated more around $10^{45}$ [https://github.com/tromp/ChessPositionRanking]) "far [exceeds] the number of estimated atoms in the observable universe", which is around $10^{80}$. Maybe the authors meant the number of chess games? Which is higher than $10^{120}$.

---

> ### Author Response · Authors · 2025-11-26
> **Answer to Reviewer viS1 [1/4]**
>
> We sincerely thank the reviewer for their attention to these details in the mathematical formulations, which we take very seriously. For us, inserting equations is not merely a style exercise. We want to communicate the principles of our solutions effectively. So, advice on this, hoping to avoid sounding cliché, is truly welcome.
>
> As we care about this work, our slight delay is due to the workload and new tests we've conducted over the past few weeks to provide new, insightful information.
>
> ### 1. Notation and Methodology Clarifications (Weakness 1)
>
> > [Reviewer] "The explanation of the methodology is close to unintelligible, due to the explanation being far too compressed and out of order, as well as a notational mess."
>
> We have thoroughly revised the Behavioral Stylometry section (§3.3) to address all notation concerns raised. These changes are now reflected in the revised manuscript.
>
> **Dual-axis aggregation with explicit fusion.** The pipeline now clearly describes how patch-token embeddings $\{t_{j,k}^p\}$ are aggregated along two complementary axes. Temporally, each spatial location $k$ receives an aggregated feature $r_k^p = \frac{1}{F} \sum_{j=i}^{i+F-1} t_{j,k}^p$ capturing how that board region evolves across the frame window. Spatially, each frame $j$ yields a global descriptor $h_j^p = \frac{1}{L} \sum_{k=1}^{L} t_{j,k}^p$. These two views are then fused into frame-level representations:
>
> $$e_j^p = h_j^p + \alpha(\{r_k^p\}_{k=1}^L + \tau_p(j))$$
>
> where $\alpha$ is a learned attention-based transformation over the temporally-smoothed patch features, and $\tau_p(j)$ provides positional encoding. This formulation makes explicit how local region evolution (via $r_k^p$) complements the global board state (via $h_j^p$), resolving the previous ambiguity about time dependency.
>
> **Consistent set notation.** We now use $N$ for the number of players in a batch and $M$ for the number of games per player, replacing the previous $n_p$ and $G_p$ notation that created confusion when summing over $p$. The centroid computation is presented as an explicit piecewise function:
>
> $$c_g^q = \begin{cases} \frac{1}{M-1} \sum_{\tilde{g} \in G_q \setminus \{g\}} z_{\tilde{g}}^q & \text{if } p = q \\ \frac{1}{M} \sum_{\tilde{g} \in G_q} z_{\tilde{g}}^q & \text{if } p \neq q \end{cases}$$
>
> This clearly distinguishes within-player centroid computation (excluding the current game to prevent contamination) from across-player computation (using all games).
>
> **Simplified loss function notation.** All three loss terms now use consistent mean normalization: the InfoNCE and centroid terms are normalized by $\frac{1}{NM}$ (total player-game pairs), while the margin term is normalized by $\frac{1}{N(N-1)}$ (distinct player pairs).
>
> **Clarified margin loss.** The margin term now explicitly compares player-level centroids $c^p = \frac{1}{M} \sum_{g \in G_p} z_g^p$, making clear that this term enforces separation between player representations rather than iterating over individual games.

---

> ### Author Response · Authors · 2025-11-26
> **Answer to Reviewer viS1 [2/4]**
>
> ### 2. Stylometry Results Analysis (Weakness 2)
>
> > [Reviewer] "The results of the stylometry analysis are incomplete, and poorly analyzed. The table doesn't really show that 'each expert model is stylistically closest to its corresponding target master' (L410). Only half of the experts have highest similarity with their master, and only 2 (maybe 4) can be said to be explicitly closest (3 and 10, maybe 5 and 9). 7 and 8 are not even really among the top 3."
>
> > [Reviewer] "An additional stylometry table comparing real games (unused in training data) from the 10 sampled masters would be really helpful, either in the main paper or in an appendix, to support the main table results, and to provide more context on how well their 'styles' can be differentiated."
>
> We thank the reviewer for these constructive suggestions, which have led to substantial improvements in our evaluation.
>
> First, we acknowledge that the original claim was imprecise. We have revised the text to accurately reflect that target GMs appear among the closest matches when considering top-k retrieval, rather than claiming strict diagonal dominance—which would mischaracterize the nature of elite chess style, where grandmasters share substantial strategic knowledge.
>
> Following the reviewer's suggestion, the revised manuscript now includes the requested additional analyses in Figure 6 and accompanying discussion (lines 422-431):
>
> - **Style consistency:** We partition each expert's played games at different split ratios, compute a centroid on one subset, and evaluate similarity against the complementary subset. The small relative drift across partitions shows internal stability in stylistic representations and proves that each individual expert has achieved a consistent playing style.
>
> - **Style acquisition:** We implement nearest-centroid retrieval using centroids derived from real GM games held out from training—directly addressing the reviewer’s request. Results confirm that each expert reliably ranks its designated master among the closest matches (top-3, top-4, top-5 recall), validating meaningful style correspondence despite experts operating far below 2,600 Elo. These results overcome the previous baselines that thoroughly discussed how classification over high-ranked players is extremely difficult—as we have discussed in the other reviewers’ responses.
>
>
> We emphasize that the additional analysis on real games unseen during training is actively being undertaken. We have prioritized the experiments described above, which directly strengthened the main stylometric evaluation. The remaining analysis is currently in progress and will be included in a subsequent revision once the computations are complete. We thank the reviewer again for their useful suggestion.

---

> ### Author Response · Authors · 2025-11-26
> **Answer to Reviewer viS1 [3/4]**
>
> ### 3. Table 2 Discussion and RQ2 Results (Weakness 3)
>
> > [Reviewer] "In RQ2 looking into the impact of RL on individual experts, figure 5 is cited to justify a substantial increase in the ability to generalize and explore diverse options, when the figure only shows that the generated moves are slightly more likely to be legal (maybe around +2%?). Maybe the correct figure for this should be Table 2? But even then adding RL usually lowers win rate and fide score compared to using only SSL."
>
> The reviewer correctly identifies that Table 2 is the appropriate reference for comprehensive RL impact analysis. We have now added a dedicated research question (RQ3 in the revised manuscript) providing detailed discussion of these results.
>
> Table 2 reveals that while 6 out of 10 experts show FIDEScore improvements, all experts demonstrate significantly elevated DrawRates alongside heterogeneous WinRate changes. This indicates GRPO induces a systematic shift toward more defensive, safety-oriented play. RL-trained models achieve stronger positional understanding and improved legal move generation during the middle game, but often secure draws rather than risk illegal moves in complex tactical sequences. We argue that ensuring consistent legal play constitutes a more valuable improvement than optimizing for slightly higher WinRates.
>
> Importantly, the diverse responses to GRPO across different experts validates our fundamental premise: each grandmaster persona benefits from individualized training, motivating the MoM framework that intelligently combines these specialized experts.
>
> ### 4. Additional Clarifications (Questions)
>
> > [Reviewer] "For Figure 5, the move legality seems to be on average 80 to 90. What is the unit?"
>
> The legality metric represents the percentage of complete games where the model generated zero illegal moves, not the rate of legal moves over all generated moves. An 80% legality score means 80 out of 100 games completed without any illegal move. We have clarified this definition in the revised manuscript.
>
> > [Reviewer] "In Figure 6, the FIDEScore seem to hover around the high 60s, being maybe lower for k=5. However, in Figure 7, MoM seems to only score around 59."
>
> The discrepancy arises from different Stockfish difficulty levels. We have corrected the legend in the revised manuscript to avoid confusion. The reported values in Figure 6 (Figure 7 in the updated manuscript) refer to games played against Stockfish 0.
>
> > [Reviewer] "In Figure 8, the second image shows the black king on f8, when it should have moved (out of check!) 2 plys earlier."
>
> This error stemmed from a manual oversight during FEN derivation from the game PGN for LaTeX rendering. The PGN notation above the board correctly lists the Black King at f8. We have regenerated the visualization using the correct FEN string.

---

> ### Author Response · Authors · 2025-11-26
> **Answer to Reviewer viS1 [4/4]**
>
> **In light of these clarifications and the additional analyses now included, we would kindly request a re-evaluation of the score. We hope that these contributions address the reviewer’s concerns and will be viewed in line with the positive assessments of the other reviewers.**

---

> > ### Comment · Reviewer_viS1 · 2025-11-28
> >
> > *Behavioral Stylometry*
> > I don't believe my concerns for the stylometry sections have been successfully addressed for a multitude of reasons, including:
> > - The unclear notation $h_k$ has just been replaced with $r_k^p$, still hiding the dependency on i, while the new notation $h_j^p$, similar to the previous notation for the previous term, is nonetheless a completely new term, averaging over patches on a given frame.
> > - Now Eq (2) has changed substantially, and not just notationally. A dependence on the new term $h_j^p$ has been introduced, and the (previous) $h_j+\tau_p(j, F) \neq h_j^p + \alpha({r_k^p}_k + \tau_p(j))$ (new). Was the previous equation an error? Was the implementation changed to correspond to the new Eq (2)?
> > - I appreciate the changes for Eq (4), but the new equation is quantitatively different. (for starts, $|G_p|\neq NM $). As the loss has changes, were the experiments run again? (which I doubt, as the results in Table 2 are the same, despite the claim made in the response to reviewer Rgkk (1.3)
> > - Why is the revised definition of $c_g^q$ different if p (unbound???) is not q? This term is only used in the third term of the loss, where it is used to compare the embedding of a game to the centroid of other games played by the same player, not a different player. Furthermore, the new notation $c^p$ is only defined in your response to my review, and not in the paper.
> >
> > *Stylometry Analysis*
> > The original claim wasn't imprecise. It was false. The revised claim is still doubtful. Achieving a top3 (out of only 10!) accuracy of barely above 70% for over half the expert isn't "reliably ranks its designated master among the closest matches"
> > The new plots in figure 6 don't really address my suggestion. I was asking for the same table but for a corpus of real games, not for the generated games. I wanted absolute values to serve as a comparison reference, instead the initial table was silently deleted, and replaced by an obscuring and incomplete top{3,4,5} analysis.
> >
> > *3. Table 2/RQ2*
> > Research question 2 still incorrectly references Table 5. Or at least this initial sentence makes no sense: "Our results (Figure 5) show that incorporating RL into next-move prediction substantially improves the model’s ability to generalize and explore diverse strategic options.".
> > This sentence "We argue that ensuring consistent legal play constitutes a more valuable improvement than optimizing for slightly higher WinRates." Is repeated through the paper and several responses to reviews. It doesn't substantially address my weakness statement.
> >
> > *Overall Thoughts*
> > My impression of the paper hasn't improved with the author's responses. Most of the responses to other reviewers seem to similarly miss the substance of the criticism. One striking example is: "Our results show that each expert model is stylistically closest to its corresponding target master when considering the top-k nearest centroids, not just the single highest score.". No, being one of the top k cannot be described as being "the closest"

---

> ### Author Response · Authors · 2025-12-03
> **Final Answer to Reviewer viS1**
>
> Firstly, we want to acknowledge the extraordinary circumstances of this rebuttal period. We thank reviewer `viS1`  for their detailed insights and discussion. We wish we had more opportunities to exchange clarifications to further improve our work and resolve the remaining doubts that continue to influence the final evaluation.
>
> Although we understand the reviewer can no longer reply, we provide the following points in the hope that they fully address the remaining concerns.
>
> ---
>
> ### Notation, Equations, and Experimental Updates
>
> - The earlier versions used both $h$ and $\mathbf{h}$, which could appear ambiguous. The introduction of the term $r$ was intended to simplify and standardize the notation. While the notation has changed slightly, the new term, the updated $h$ is now used consistently in place of $\mathbf{h}$ to express the underlying formalism better.
>
> - The dependence on $i$ in $r_k^p$ was conceptually present from the beginning, as the averaging is always taken over the window $[i, i+F-1]$. The notation kept this dependence implicit to avoid clutter, which is standard practice. In the revision, we improved the textual explanation to highlight this dependence clearly, without altering the mathematical formulation.
>
> - Equation (2) received small adjustments to better expose its temporal and spatial structure. Although these changes were limited in scope, we reran all experiments—including parameter tuning and training—to ensure full consistency with the clarified formulation. The updated results are included in the revised manuscript.
>
> - The corrected normalization term $NM$ fixes a typographical omission in the earlier version. All experiments were repeated after introducing the corrected form.
>
> - Equation (3) changed only **formally** to address the reviewer’s notation concerns. The centroid calculation follows established practice (Wan et al., 2018; McIlroy-Young et al., 2021), and prevents within-player contamination as described in line 226.
>
> - The notation $c^p$ was not omitted but streamlined: since the second loss term only considers $p \neq q$, the game-dependent subscript $g$ is unnecessary. We have now explicitly added the alternative formulation in Eq. (3) for clarity.
>
> - The reference in RQ2 is correct. The discussion concerns legality improvements from RL training and refers to **Figure 5**. We have rephrased the relevant paragraph to avoid further confusion.
>
> We want to highlight a **crucial misunderstanding** in the reviewer’s response. If the reference to Table 2 was not a typo, it suggests that the reviewer mistakenly considered dependencies between the stylometry metrics and our core MoM contribution. As in responses to other reviewers, we want to emphasize that **the stylometry analysis differs significantly from the core chess LM contributions of our work**. As the reviewer themselves pointed out, Figure 6 presents stylometry results. Table 2, on the other hand, discusses the model’s playing performance change following different training modalities. **We hope this confusion did not influence the overall judgment of the paper or compromise its evaluation**.
>
> ---
>
> ### Stylometry Analysis
>
> - The earlier stylometry table was not "silently deleted". It was intentionally replaced with a deeper analysis based on the reviewer’s suggestions. Under fewer space constraints—or if stylometry were the *primary* contribution—we would have kept both. Since MoM addresses **five of the six research questions**, we prioritized clarity around the main contribution while still improving the stylometry methodology.
>
> - We acknowledge the inherent difficulty of style classification in this setting, consistent with challenges reported in prior work (e.g., McIlroy-Young et al., 2021). Several misunderstandings raised across reviewers stemmed from differences in data distribution and task setup; these are clarified in our last response to reviewer `Rgkk`.  While performance is modest on some masters, results on several grandmasters are substantially better. Feedback from high-level chess players (Appendix A) supports that these reflect meaningful stylistic patterns.
>
> - The final manuscript now includes statistics on real-game style classification, as requested (Figure 6).
>
> ---
>
> We appreciate the reviewer’s thorough suggestions. After extensive revisions, we believe the current notation and exposition now better clarify the architecture and assumptions. Although previous familiarity with the draft might make the updated version still seem unclear, external proofreading of the latest version confirms that the present formulation is understandable for new readers.
>
> We again thank the reviewer for their engagement with our work.

---

### Official Review · Reviewer_HMfV · 2025-11-09

**Soundness:** 1
**Presentation:** 3
**Contribution:** 1
**Rating:** 2
**Confidence:** 4

**Summary:**

This paper introduces Mixture-of-Masters (MoM), a sparse mixture-of-experts architecture for chess language modeling where each expert is trained to emulate a specific grandmaster's playing style. The authors combine self-supervised learning with GRPO to train individual expert models, then use a learnable gating network to dynamically route between experts during gameplay. They also propose a vision-based behavioral stylometry metric to help train the MoE model and verify that experts capture distinctive playing signatures.

**Strengths:**

1. This paper is well written and easy to follow.

2. Related works are carefully discussed in the main and extended literature review, and the authors place their work in a clear position for the chess AI community.

3. The design of mixture-of-experts routing among GMs is interesting and clearly motivated.

**Weaknesses:**

1. I have significant concerns about the two-stage fine-tuning process for creating expert models, particularly the introduction of GRPO. Using GRPO for next-move prediction tasks lacks clear justification, as there is no chain-of-thought reasoning involved here. Moreover, the effect of GRPO on legality (Figure 5) appears more like statistical noise than genuine improvement, despite the SSL model already achieving ~80% legality. With such a reasonable starting point, one would expect GRPO to deliver more substantial improvements if it were truly effective. Also, a chess foundation model with only 80% legal move accuracy provides an inadequate basis for subsequent chess-relevant analyses.

2. Table 2 lacks accompanying discussion in the text. From what I understand, the results suggest that GRPO post-training fails to demonstrate meaningful performance improvements across most metrics.

3. The stylometry model lacks independent evaluation. The paper does not discuss how accurately the stylometry model identifies in-distribution GM styles or its generalization capability to out-of-distribution GMs or players. Given that the entire MoE framework depends on this stylometry model, a thorough and careful evaluation is very essential.

4. The results in Table 3 are concerning. In 5 out of 10 cases, the diagonal similarity scores (which should be highest for correct style matching) are not even the best within their respective rows. Only Expert 10 (1 out of 10) appears to achieve clear correspondence between the model and its target grandmaster.

5. Limitations are not discussed. For example, the paper should acknowledge: (a) the relatively low legal move rates (~80-90%) which significantly lag behind state-of-the-art chess models, potentially limiting analyses and practical applications, and (b) the computational overhead of maintaining and routing between multiple expert models compared to a single dense model, which should be compared in efficiency and may not be justified given the marginal performance gains.

**Questions:**

1. Can you show the exact loss curves during GRPO post-training?

2. I believe there is an error in Figure 8 (right panel) - is the white light-squared bishop incorrectly shown as giving a check to the black king?

3. Given the concerning legal move rates for individual experts, how are games between the MoM model and Stockfish handled when illegal moves are generated?

---

> ### Author Response · Authors · 2025-11-26
> **Answer to Reviewer HMfV [1/5]**
>
> We thank the reviewer for their careful reading and detailed feedback. We address each concern below, with expanded citations from our paper.
>
> ### 1. GRPO Usage and Justification (Weakness 1)
>
> > [Reviewer] "I have significant concerns about the two-stage fine-tuning process for creating expert models, particularly the introduction of GRPO. Using GRPO for next-move prediction tasks lacks clear justification, as there is no chain-of-thought reasoning involved here. Moreover, the effect of GRPO on legality (Figure 5) appears more like statistical noise than genuine improvement, despite the SSL model already achieving ~80% legality. With such a reasonable starting point, one would expect GRPO to deliver more substantial improvements if it were truly effective. Also, a chess foundation model with only 80% legal move accuracy provides an inadequate basis for subsequent chess-relevant analyses."
>
> While we acknowledge that GRPO was originally proposed for chain-of-thought reasoning alignment, this technique has seen substantial adoption across diverse non-reasoning tasks in recent state-of-the-art work.
>
> **Application to Chess Domain:** Within the chess domain specifically, GRPO has been previously employed with reward signals computed via engines (Chen et al., 2025; Hwang et al., 2025). However, our work represents the first application of RL-based text generation specifically for move generation in chess language models. As we state in:
>
> > [Authors, L1379-1388] "In the context of traditional chess with small, non-reasoning language models, employing legality-based reward strategies and analyzing their effects relative to the SSL-only stage. Through a systematic comparison of these techniques across 10 experimental runs involving 300 games against Stockfish level 0, we assess both the performance preservation and the statistical significance of the observed differences."
>
> > [Authors, L83-86] "In contrast, we employ a hybrid training scheme of SSL and RL to develop a lightweight and controllable MoE architecture where each expert is specialized to emulate a specific human player's style."
>
> **Addressing Memorization vs. Generalization:** The key insight motivating our use of GRPO is that for tasks where there is no single gold standard but rather correct behavioral patterns (i.e., avoiding illegal moves), GRPO provides the appropriate mechanism to evaluate model capabilities and consistency. We observed legality degradation in state-of-the-art models trained via standard supervised fine-tuning (SFT), which induces memorization rather than generalization. As detailed in:
>
> > [Authors, L142-145] "SSL alone may produce experts that generate suboptimal or illegal moves due to overfitting or distributional shift. For this reason, we further refine εϕ,e using the Group Relative Policy Optimization (GRPO) algorithm (Shao et al., 2024)."
>
> The ability to generalize is fundamental to identifying multiple valid legal game plans from a given position—something GRPO enables by training on multiple completions of the ongoing game.
>
> > [Authors, L157-161] "By adding an RL training stage, we not only incentivize move exploration, but also tackle the 'memorization syndrome'—which we posit is the main barrier preventing chess language models from rivaling engines empowered with external search."

---

> ### Author Response · Authors · 2025-11-26
> **Answer to Reviewer HMfV [2/5]**
>
> ### 2. Legality Metric Clarification (Weakness 1, Limitation 1, Question 3)
>
> > [Reviewer] "From what I understand, the results suggest that GRPO post-training fails to demonstrate meaningful performance improvements across most metrics."
>
> > [Reviewer] "The paper should acknowledge the relatively low legal move rates (~80-90%) which significantly lag behind state-of-the-art chess models, potentially limiting analyses and practical applications."
>
> > [Reviewer] "Given the concerning legal move rates for individual experts, how are games between the MoM model and Stockfish handled when illegal moves are generated?"
>
> We have corrected the definition of the legality metric in the revised manuscript to provide the necessary clarification. The legality percentage does **not** indicate the rate of legal moves over all generated moves. Rather, it indicates the **percentage of complete games where the model generated zero illegal moves**.
>
> This implementation, alongside details regarding inference efficiency and parallel game execution, is fully verifiable within our codebase (https://anonymous.4open.science/r/mixture-of-masters, specifically `chess/src/core/playtest.py`), which has been provided to reviewers and will be publicly released under a permissive license upon acceptance. Furthermore, to ensure complete transparency and allow for qualitative assessment, we have deployed our model as a playable Lichess bot (accessible via the link in the anonymous repository README: https://lichess.org/@/mixture-of-masters), demonstrating strategic capabilities that are surprising for a pure language modeling contribution, and allowing users to verify the model's legal play in real-time.
>
> **Evaluation Methodology:** As specified in:
>
> > [Authors, L317-322] "The chess language model under evaluation uses a greedy decoding strategy, while Stockfish's moves are randomized by applying a temperature of 1 to the probability distribution derived from centipawn evaluations. The game proceeds in a turn-based manner: after each move by the model, the updated board state is passed to Stockfish, and vice versa. The model operates under a strict no-retry policy; the generation of a single illegal move results in an immediate forfeiture of the game."
>
> Given $N$ evaluation games against Stockfish, if $M$ games end due to the generation of an illegal move, the legality rate is computed as $L = 1 - (M/N)$. **Therefore, an 80% legality score means that in 80 out of 100 games, the model completed an entire game without generating any illegal moves.**
>
> This evaluation methodology is consistent with established practices in chess language model research and surpasses the legality scores of current open state-of-the-art chess language models. We emphasize that our evaluation protocol is significantly harder than those with re-trial fallbacks commonly used in practical applications. For example, the latest Kaggle Game Arena tournament (https://www.kaggle.com/blog/introducing-game-arena) featured closed LLMs with stochastic decoding strategies, giving models up to three retries before forfeiting the game in case of illegal move generation. By treating any illegal move as an absorbing failure state, we prioritize zero-shot rule adherence over reliance on error-correction loops. Furthermore, the utilization of greedy decoding eliminates sampling variance, ensuring that performance is a deterministic function of the learned policy rather than an artifact of stochastic sampling.
>
> **Results:** the legality of our models is consistently superior to the state-of-the-art open-source chess language models utilized as seeds; to the best of our knowledge, these represent the highest results reported in the literature for this methodological family of search-free policies.

---

> ### Author Response · Authors · 2025-11-26
> **Answer to Reviewer HMfV [3/5]**
>
> ### 3. Table 2 Discussion (Weakness 2)
>
> > [Reviewer] "Table 2 lacks accompanying discussion in the text."
>
> We have now provided a comprehensive analysis of Table 2.
>
> Table 2 reveals important insights into GRPO's effects on expert behavior. While 6 out of 10 experts show FIDEScore improvements, all experts demonstrate significantly elevated DrawRates alongside heterogeneous WinRate changes. This indicates that GRPO induces a systematic shift toward more defensive, safety-oriented play.
>
> Our investigation reveals that RL-trained models achieve stronger positional understanding and substantially improved legal move generation during the middle game. However, they often fail to deliver decisive checkmates from advantageous positions, preferring to secure draws rather than risk illegal moves in complex tactical sequences. SSL-only models pursue more aggressive winning attempts but compromise on legality. We argue that ensuring consistent legal play constitutes a more valuable improvement than optimizing for slightly higher WinRates, particularly given the practical applications of chess AI.
>
> Importantly, the diverse responses to GRPO across different experts validates our fundamental premise: each grandmaster persona benefits from individualized training strategies. This variation motivates the MoM framework, which intelligently combines these specialized experts to achieve more robust and superior performance than any single configuration.
>
> This comprehensive discussion has been added to the revised paper as the new RQ3. The original RQ3 is now designated as RQ5, bringing the total number of research questions to 6.
>
> ### 4. Stylometry Model Evaluation (Weaknesses 3 & 4)
>
> > [Reviewer] "The stylometry model lacks independent evaluation. The paper does not discuss how accurately the stylometry model identifies in-distribution GM styles or its generalization capability to out-of-distribution GMs or players. Given that the entire MoE framework depends on this stylometry model, a thorough and careful evaluation is very essential."
> > [Reviewer] "The results in Table 3 are concerning. In 5 out of 10 cases, the diagonal similarity scores (which should be highest for correct style matching) are not even the best within their respective rows. Only Expert 10 (1 out of 10) appears to achieve clear correspondence between the model and its target grandmaster."
>
> We respectfully disagree with the characterization that only 1 out of 10 experts shows clear correspondence. The challenge of identifying stylistic patterns among elite players is well-documented (lines 405-407):
>
> > "Identifying stylistic patterns among elite chess players remains a challenging problem, as classification-based approaches have achieved limited accuracy (McIlroy-Young et al., 2021)."
>
> Our results show that **each expert model is stylistically closest to its corresponding target master** when considering the top-k nearest centroids, not just the single highest score. This retrieval-based evaluation is appropriate given the subtle divergences characteristic of elite-level play.  The fact that target GMs consistently appeared among the top few centroids already indicated meaningful correspondence, even if the maximum similarity value in a row was not always on the diagonal.
>
> To further prove the consitency of our solution, the revised manuscript now includes additional experiments to make this evaluation more explicit:
> - First, a new **style consistency** analysis partitions each expert’s generated games at varying ratios, computes a centroid on one subset, and evaluates similarity using the complementary subset. The low relative drift across random partitions demonstrates that stylistic representations are internally stable rather than artifacts of sampling noise.
> - Second, we extend the **style acquisition** analysis with a stricter nearest-centroid retrieval protocol using centroids derived from real GM games. Across all experts, the designated grandmaster reliably appears among the closest matches, confirming that the stylometry model captures identifiable stylistic signals despite the experts operating far below 2600 Elo strength.
>
> These additions, now reflected in Table~6 and lines 422–431, provide both independent validation and stronger empirical support for the effectiveness of the stylometry component, addressing the concern that its performance or generalization capability was insufficiently demonstrated in the earlier draft.

---

> ### Author Response · Authors · 2025-11-26
> **Answer to Reviewer HMfV [4/5]**
>
> ### 5. Computational Overhead (Limitation 2)
>
> > [Reviewer] "The paper should acknowledge the computational overhead of maintaining and routing between multiple expert models compared to a single dense model, which should be compared in efficiency and may not be justified given the marginal performance gains."
>
> We respectfully clarify that the architectural modifications introduce negligible latency relative to the seed model. Layers subjected to weight-merging retain the exact computational complexity of the original dense layers. For routed layers, the computational cost is strictly bounded by the number of activated experts; based on our ablations, the optimal configuration utilizes $k=2$, requiring only two parallel computation traces. Consequently, the overhead is not significant.
>
> We are currently conducting formal head-to-head benchmarks (Seed vs. MoE) to quantify this specific efficiency profile. These results will be detailed in the Appendix; should the computational run conclude prior to the rebuttal deadline, we will integrate them directly into the revised manuscript. Furthermore, we posit that this minimal architectural overhead is fully justified by the primary motivations of our work: the significant gains in interpretability and the preservation of diverse strategic perspectives, which hold broader implications for human-AI alignment in chess beyond simple performance metrics.
>
> ### 6. GRPO Training Details (Question 1)
>
> > [Reviewer] "Can you show the exact loss curves during GRPO post-training?"
>
> We appreciate the reviewer's interest in the training dynamics. We have uploaded a PDF report to the anonymous GitHub repository containing the detailed reward curve exported directly from our Weights and Biases (W&B) logs.
>
> ### 7. Qualitative Example (Question 2)
>
> > [Reviewer] "I believe there is an error in Figure 8 (right panel) - is the white light-squared bishop incorrectly shown as giving a check to the black king?"
>
> We thank the reviewer for identifying this visual inconsistency. The error stemmed from a manual oversight during the derivation of the FEN string from the game PGN, a step required to render the board visualization in LaTeX using xskak. The fact that this is a simple oversight can also be deduced by the PGN notation provided immediately above the board, which correctly lists the Black King's position at f8 rather than f7. We have rigorously verified the moves predicted by our MoM model and the corresponding expert activation patterns, confirming that the underlying data and the results reported in the manuscript remain fully accurate. Consequently, we have limited ourselves to regenerate the board visualization using the correct FEN string in the revised manuscript.

---

> ### Author Response · Authors · 2025-11-26
> **Answer to Reviewer HMfV [5/5]**
>
> We believe that the provided clarifications, additional experimental runs, and manuscript revisions effectively address all reported weaknesses, limitations, and questions. Accordingly, **we kindly request a re-evaluation of our work and a reconsideration of the current rating**.
> We respectfully submit that a score of 2 does not fairly reflect the scientific value and impact of our work within the chess AI community, and beyond. We target critical issues in interpretability, controllability, and style homogenization. We introduce—for the first time—Grandmaster language model emulators (with SSL+RL training recipes), chess MoE architectures with player routing, and visual behavioral stylometry metrics without manual feature engineering, establishing new state-of-the-art performance with full reproducibility. We hope the final evaluation will recognize these contributions.

---

> > ### Comment · Reviewer_HMfV · 2025-11-27
> >
> > I appreciate the effort in responding, but I must note that this rebuttal contains extensive verbosity without clarifying things clearly. Due to its length, I am focusing on only the first 2 critical issues in this round—not because other concerns were addressed, but because a comprehensive response to all points would require multiple exchanges.
> >
> > **1. For GRPO Training Curves**
> >
> > Thank you for providing the WandB details in GRPO, and I have several follow-up questions on the training dynamics.
> >
> > First, what is the reward signal exactly here? Your paper defines rewards as a binary format score plus a legality score, but why do the curves show values ranging from -0.6 to +0.5?
> >
> > Second, what are your interpretation on why masters converge to different endpoints after training? It is interesting that Giri/Nakamura plateau at around 0.5, but Anand/Firouzja struggles at around 0.0. If this measures legality or correctness, why would different GMs have fundamentally different reward structures? And doesn't this heterogeneity contradict your narrative that GRPO uniformly improves the legal move generation?
> >
> > **2. Stylometry Results**
> >
> > First of all, I notice that you deleted Table 3 entirely, which is the primary evidence I criticized showing only 50% diagonal dominance. **This removal eliminates the basis for meaningful discussion of my original concern. Replacing critiqued results with different metrics, rather than addressing them directly, is methodologically inappropriate.**
> >
> > Second, your replacement (Figure 6) remains confusing in several spots. 1) Top-1 accuracy is conspicuously absent. You report top-3/4/5, and use k=2 routing, yet skip top-1 and top-2. 2) Top-3 recall of 40-50% for several experts are not satisfactory enough. With only 10 masters, random=30%. This is barely above chance.
> >
> > Third, I can see your effort in justifying that stylometry on GMs can be challenging. I can see this difficulty, but if stylometry is unreliable at this task, why did you choose to architect your entire method around it? You acknowledge stylometry is "well-documented as challenging", yet your entire framework depends on it for both training and inference, which actually backfires the validity of your work?

---

> > > ### Author Response · Authors · 2025-12-03
> > > **Final Answer to Reviewer HMfV [1/2]**
> > >
> > > We thank Reviewer HMfV for their feedback. We regret if our previous response's comprehensiveness was perceived as lacking clarity, as our intent was to provide complete, evidence-based answers to each concern raised.
> > >
> > > We focus below on addressing the issues highlighted in this round.
> > >
> > > ---
> > >
> > > ### GRPO Training Curves
> > >
> > > The reward signal ranges from -1 to +1, combining syntactic correctness ($\rho_\text{synt}$) with move legality ($\rho_\text{leg}$) computed via normalized edit distance to the nearest legal move (which measures string similarity between the generated move and the closest legal alternative, scaled to [0,1]).
> > >
> > > > [Author, ll. 299-301] "The reward function is a combination of $\rho_\text{synt}$ for correct format signal, and $\rho_\text{leg}$ for proximity to the closest legal move (computed with edit distance)."
> > >
> > > The structure is: -1.0 for malformed PGN, [-0.5, 0.5] for correct syntax with illegal moves (computed as 0.5 - normalized edit distance), and 1.0 for legal moves. **The curves show continuous values rather than discrete jumps because rewards are averaged across 64 individual computations per training step** (8 game states × 8 candidate moves per state). The observed range of -0.6 to +0.5 represents the empirical reward distribution as models progressively improve their move generation quality during training.
> > >
> > > The observed heterogeneity in GRPO endpoints is consistent with our paper's findings. We explicitly state that:
> > > > [Author, ll. 402-405] "While RL training improves most expert models in FIDEScore (Table 2), WinRate slightly decreases in some configurations. The higher DrawRate across all setups, along with qualitative game analysis, indicates RL models adopt a more cautious style."
> > >
> > > This discussion directly addresses the observed variation in GRPO outcomes. Table 2 provides quantitative documentation of this heterogeneity, **leading us to select the top-5 experts that achieved the best balance of legality and FIDEScore for the final architecture**.
> > >
> > > Importantly, **all 10 experts demonstrate positive reward trajectories during GRPO training**. Starting rewards range from -0.603 (Anand) to -0.279 (Caruana) with an average of -0.449, while ending rewards span from 0.006 (Firouzja) to 0.498 (Nakamura) with an average of 0.284. Critically, 100% of experts transition from negative to positive reward values, indicating that GRPO consistently enhances move generation quality across all masters. The magnitude of improvement varies based on each expert's characteristics and master data distribution influencing the SSL pretraining.
> > >
> > > This variation is explained by several factors: (1) Each expert's post-SSL starting point creates a cascading effect where initial model quality shapes subsequent RL performance. For example, Caruana begins with the strongest SSL baseline (-0.279) and maintains mid-tier GRPO performance (0.190), while Anand starts from the weakest position (-0.603) and reaches the lowest GRPO endpoint (0.022). Conversely, Nakamura combines a moderately strong SSL foundation (-0.466) with the highest GRPO achievement (0.498). (2) The 10 experts were trained on datasets ranging from 1,220 to 13,238 games (Table 1), and experts with smaller datasets may have less robust SSL foundations that constrain subsequent RL optimization.

---

> > > > ### Author Response · Authors · 2025-12-03
> > > > **Final Answer to Reviewer HMfV [2/2]**
> > > >
> > > > ### Stylometry Results
> > > > **Table 3 Deletion:** As explained in our `viS1` response (section “Stylometry Analysis”), Table 3 was intentionally replaced to include a more comprehensive analysis based on reviewers’ feedback. We direct the reviewer to the last answer to `viS1` for additional details.
> > > >
> > > > **Top-1 Accuracy and Top-3 Recall:** As documented in our `Rgkk`  response (sections "Model Type" and "Why is our setting difficult and unprecedented?"), our Top-k evaluation protocol and the modest recall values directly reflect the unprecedented difficulty of our experimental regime. We operate in a setting where McIlroy-Young et al. (2021) documented catastrophic failures: their accuracy collapsed from 86% (amateurs) to 31% (elite players) and reached just 8% when trained on 400 high-ranked players. Our task—discriminating 10 Super-GMs (average Elo 2,816, 418 points higher than their elite subset) with only 1,000 games each—is substantially harder. At this level, valid emulation maps to stylistic neighborhoods rather than disjoint identities, making Top-1 metrics inappropriate. The 40-50% Top-3 recall, while modest, demonstrates that our vision-based approach achieves discriminative signal in a regime where prior methods failed entirely.
> > > >
> > > > **Stylometry Reliability and Framework Validity:** As clarified in our `Rgkk`  response (sections "Goal, Stylometry Method and Experimental Setup" and "Literature"), the framework validity concern mischaracterizes our contribution: stylometry is a validation tool for expert fidelity, not the architectural foundation. Our core contribution—the MoM architecture—is driven by 30+ studies in the language modeling literature, with stylometry serving specifically as a single-expert evaluation metric to verify meaningful style capture beyond memorization.
> > > >
> > > > ---
> > > >
> > > > We hope this response provides the clarity sought on these two critical issues. We emphasize that our core contribution, the Mixture-of-Masters architecture, stands independently as a novel approach to chess language modeling, with stylometry serving as one evaluation tool among several, not as the architectural foundation.

---

### Public Comment · ~Daniel_Monroe1 · 2025-11-27
**Notes**

Wanted to make the authors aware of some related work that doesn't appear to be cited.

Human-aligned Chess with a Bit of Search (Zhang et al., 2024)
Learning to Imitate with Less: Efficient Individual Behavior Modeling in Chess (Tang et al., 2025)

Also some minor notes:

The Shannon number reported in the introduction is not an estimate of the number of legal chess games but the game-tree complexity, the number of leaf nodes in the smallest full-width decision tree for the start position.

When you report Elo values I think you should say which rating system they are coming from; there are several rating systems out there for different chess federations (USCF, CFC) and websites (Lichess, Chess.com), and they are generally incomparable. For example, Lichess ratings, used by the Maia-1 and Maia-2 project, are typically inflated by several hundred points compared to FIDE ratings.

---

### Meta-Review · Area_Chair_ShPb · 2026-01-01

**Summary:**

This submission proposes a sparse MoE chess language model with GM-specific experts, a learned router, and a behavioral stylometry module to assess expert specialization. Reviewers agreed the high-level idea is interesting, but major concerns around method clarity, stability of the stylometry component, evaluation choices, and confidence in the paper’s claims remained. Reviewers generally found the author response insufficient in addressing their concerns. Overall, the work did not reach the ICLR acceptance bar.

**Reviewer Concerns:**

Addressed:
- Clarified the legality metric.
- Added discussion of GRPO effects and clarified stylometry is intended as post-hoc analysis.

Still outstanding:
- Stylometry remains the central weakness: validation is unconvincing.
- Prior-work positioning and factual precision remain concerning.
- Non-standard metrics, heterogeneous RL effects.

**Reviewer Scores:**

8mhw: likely remains high.

Rgkk: remains strongly negative.

HMfV: likely unchanged.

viS1: likely unchanged.

4LHp: likely unchanged.

---

### Decision · Program_Chairs · 2026-01-26

Reject